# Stochasticity, determinism, and contingency shape genome evolution of endosymbiotic bacteria

Bret M. Boyd [1,5] ✉, Ian James [2,5], Kevin P. Johnson [3], Robert B. Weiss [4], Sarah E. Bush [2], Dale H. Clayton [2] & Colin Dale[2]

Evolution results from the interaction of stochastic and deterministic processes that create a web of historical contingency, shaping gene content and organismal function. To understand the scope of this interaction, we examine the relative contributions of stochasticity, determinism, and contingency in shaping gene inactivation in 34 lineages of endosymbiotic bacteria, *Sodalis*, found in parasitic lice, *Columbicola*, that are independently undergoing genome degeneration. Here we show that the process of genome degeneration in this system is largely deterministic: genes involved in amino acid biosynthesis are lost while those involved in providing B-vitamins to the host are retained. In contrast, many genes encoding redundant functions, including components of the respiratory chain and DNA repair pathways, are subject to stochastic loss, yielding historical contingencies that constrain subsequent losses. Thus, while selection results in functional convergence between symbiont lineages, stochastic mutations initiate distinct evolutionary trajectories, generating diverse gene inventories that lack the functional redundancy typically found in free-living relatives.

Eukaryotes have benefited greatly from the acquisition of mutualistic microbial symbionts that catalyzed their diversification[1]. As a defining example, the eukaryotic cell originated as a consequence of the acquisition of an alpha-proteobacterial symbiont facilitating aerobic energy generation that subsequently became the mitochondrion[2]. Similarly, the ancestor of algae and plants acquired a cyanobacterial symbiont, which became the chloroplast[2], facilitating oxygenic photosynthesis. Many animals have evolved additional mutualisms with other prokaryotes and fungi[1,3–6]. These relationships facilitated the acquisition of novel adaptive features that propelled host diversification[3,7].

Based on the current understanding of the extent and diversity of mutualistic symbioses between microbes and their hosts, it is apparent that some eukaryotes acquired additional microbial symbionts that, like mitochondria, are intracellular. These endosymbionts are often

required for host survival and promote host diversification[3,7,8]. Endosymbionts can facilitate host diversification through the synthesis of essential metabolites, allowing the host to exploit a nutritionally incomplete diet. These metabolites enable the transition to a novel diet, minimizing competition between hosts[1,6,9]. For example, human lice consume only blood, which is replete with amino acids but lacking in essential vitamins, which are provided by their bacterial endosymbionts[5,10]. Often, these intimate associations between insects and microbes are maintained over long periods of evolutionary time, predominantly by maternal endosymbiont transmission[3,5,8,11,12] resulting in host-symbiont coadaptation[8].

As described above, insects serve as excellent models for the study of the evolution and function of mutualistic endosymbionts, because many different insect taxa have evolved intimate and obligate associations with microbes over the course of their 400-million-year

[1]Center for Biological Data Science, Virginia Commonwealth University, Richmond, VA, US. [2]School of Biological Sciences, University of Utah, Salt Lake City, UT, US. [3]Illinois Natural History Survey, Prairie Research Institute, University of Illinois, Champaign, IL, US. [4]Department of Human Genetics, University of Utah, Salt Lake City, UT, US. [5]These authors contributed equally: Bret M. Boyd, Ian James. ✉e-mail: boydbm@vcu.edu

history[13]. For example, the aforementioned relationship between human lice and their endosymbionts is estimated to be 13-25 million years old[12,14], while other nutritional symbioses are estimated to have persisted for up to 270 million years[3,15]. In contrast, some mutualistic insect-microbe associations have arisen recently, in some cases as a consequence of anthropogenic change[16,17]. Moreover, the simplicity of insect-endosymbiont associations that involve only one or two microbial partners makes them relatively easy to study using both ecological and laboratory-based approaches, including 'omics approaches[9,18]. For these reasons, the study of intimate (maternally transmitted) insect-bacterial endosymbioses can yield detailed insight into the mechanics and consequences of microbes coevolving in mutualistic relationships with eukaryotic hosts over a macroevolutionary timescale[19].

When microbes undergo lifestyle transition to obligate host-association, their genomes are often reduced in size, consistent with the loss of genes that were adaptive for their free-living ancestors, but dispensable in symbiosis[8]. Such degenerative evolution is thought to be exacerbated by (i) isolation within host cells and tissues, preventing genetic exchange, (ii) frequent population bottlenecks during vertical transovarial transmission, reducing the efficiency of selection, and (iii) the loss of DNA recombination and repair systems, increasing genetic load[20]. Consequently, long-established insect endosymbionts are known to have tiny genomes, often maintaining the minimal level of functionality required to serve their hosts[20]. Studies of more recently-derived (nascent) endosymbionts demonstrate that genome degeneration occurs rapidly after the establishment of endosymbiosis and is characterized by gene-inactivating mutations, followed by loss of pseudogenes via deletions[17].

While mutualistic endosymbionts are phylogenetically diverse, certain genera (*e.g. Sodalis* and *Arsenophonus*) are found repeatedly in a wide range of insect taxa, participating in distinct roles, suggesting that they are predisposed towards establishing associations with insects[4,9]. In the current study, we use phylogenomic and comparative genomic approaches to study the repeated evolution of *Sodalis* (Enterobacterales: Bruguierivoracaceae) endosymbionts in feather-feeding bird lice, focusing on dove lice in the genus *Columbicola* (Phthiraptera: Ischnocera). This system provides a unique opportunity to study the evolution of endosymbiont genomes because these endosymbionts originated repeatedly from closely related progenitors with similar gene inventories[21]. Following establishment of host associations, these bacteria converge on mutualistic function because all species of the host genus *Columbicola* consume feathers, which are mainly composed of keratin, a nutritionally incomplete diet[22]. These associations are shielded from environmental variation because lice are life-long ectoparasites of birds[23]. Thus, the repeated endosymbiont acquisitions in lice represent effective ecological replicates, analogous to the "replays of the tape of life" popularized by Stephen Jay Gould[24,25]. Comparative analysis of genome evolution in these endosymbionts provides a unique opportunity to gain detailed insight into the dynamic evolutionary processes underpinning symbiosis. We use a range of comparative approaches to explore the relative contributions of stochasticity, determinism, and contingency in shaping gene loss over macroevolutionary time[24,26]. Here we show that the process of genome degeneration is largely deterministic, yet stochastic gene losses lead to historical contingencies.

## Results

### Repeated Acquisition of Heritable-intracellular Bacteria
A previous study[21] showed that endosymbionts of lice were acquired repeatedly from closely related free-living ancestors. In the current study, a broad phylogenetic survey using local assembly methods, combined with high resolution phylogenomics using whole genomes, yielded compelling evidence of repeated independent acquisition of

*Sodalis*-allied endosymbionts, as well as three additional bacterial lineages, among feather-feeding lice.

Examining phylogenetic patterns among the endosymbionts of feather lice broadly, we identified multiple lines of evidence supporting the independent acquisition of endosymbionts. First, louse endosymbionts represent a polyphyletic group, composed of species belonging to four different proteobacterial lineages (Fig. 1a), requiring at least four different acquisition events by lice. Second, closely related species of lice harbor endosymbionts allied to different and distinct proteobacterial clades (Figs. S1–S4). Third, distantly related lice (*i.e.* different genera) harbor endosymbionts from the same clade (Fig. S1). These patterns imply that lice repeatedly acquired endosymbionts from environmental progenitors.

When examining only lice in the genus *Columbicola* and their associated *Sodalis*-allied endosymbionts using phylogenomic methods (inference based on 977 and 297 genes respectively), we found strong evidence for extensive independent and repeated acquisition events. The *Sodalis* endosymbiont tree is comb-like, with little internal structure and long terminal branches (Fig. 1a, b). Furthermore, the terminal branch leading to a closely related free-living species, *Sodalis praecaptivus*, is extremely short when compared to branches leading to endosymbionts (Fig. 1b, c). This tree topology is known to arise in simulations[21] under conditions in which multiple host taxa independently acquire endosymbionts that subsequently evolve at an accelerated rate, as often observed in natural systems[9]. In addition, endosymbionts of feather lice are vertically inherited[24], a process that should result in host-endosymbiont cospeciation yielding congruent host and endosymbiont phylogenetic trees (e.g. Boyd et al. [12,27]). However, a focused comparison of host (*Columbicola*) and endosymbiont (*Sodalis*) trees revealed relatively few of these events among the taxa sampled in our study (Fig. 1b). Furthermore, the few host-endosymbiont cospeciation events identified are all associated with recent host speciation events, when compared to the initial radiation of these lice (15 or 24 million years ago; Fig. 1b)[28]. This pattern suggests that endosymbiont replacement may occur frequently enough to erase evidence of past louse-endosymbiont cospeciation events. Serendipitously, this means that most lice in our study have independently acquired *Sodalis*-allied endosymbionts, facilitating the aforementioned study of "replays", in this case involving "the tape of *symbiosis*".

### Pace and drivers of genome decay
We developed a novel pipeline to assemble and annotate the genomes of feather louse endosymbionts, facilitating prediction of intact genes and pseudogenes through comparison with the free-living *S. praecaptivus*. This pipeline was validated in a comparative manner with de novo metagenomic assembly and annotation (methods available in the Supplementary Materials) of three louse endosymbiont genomes that have the highest diversity in genome size and divergence from *S. praecaptivus* (endosymbionts of *C. claytoni* and *C. rodmani*) and the lowest fraction of symbiont to host reads (endosymbiont of *C. fradei*; specific results available in the Supplementary Materials). Here we found general agreement between methods, with >99% of genes identified as intact in our novel pipeline also being identified as intact in the metagenomic assemblies. Furthermore, very few intact genes were identified in the metagenome assemblies, for which we could not identify any orthologs in the *S. praecaptivus* genome, indicating that the gene inventories of the *C. claytoni, C. fradei* and *C. rodmani* endosymbionts are near-perfect subsets of *S. praecaptivu*s.

We found that G + C content, genome size, and the count of protein coding genes (both intact and inactivated) were negatively correlated with patristic distance (evolutionary distance between two species in a phylogenetic tree) between each *Sodalis* endosymbiont and the free-living *S. praecaptivus* (Fig. 2). Visualization of the complete louse endosymbiont genome annotation (Fig. 3) reveals that genome degeneration in the louse *Sodalis* endosymbionts occurred in

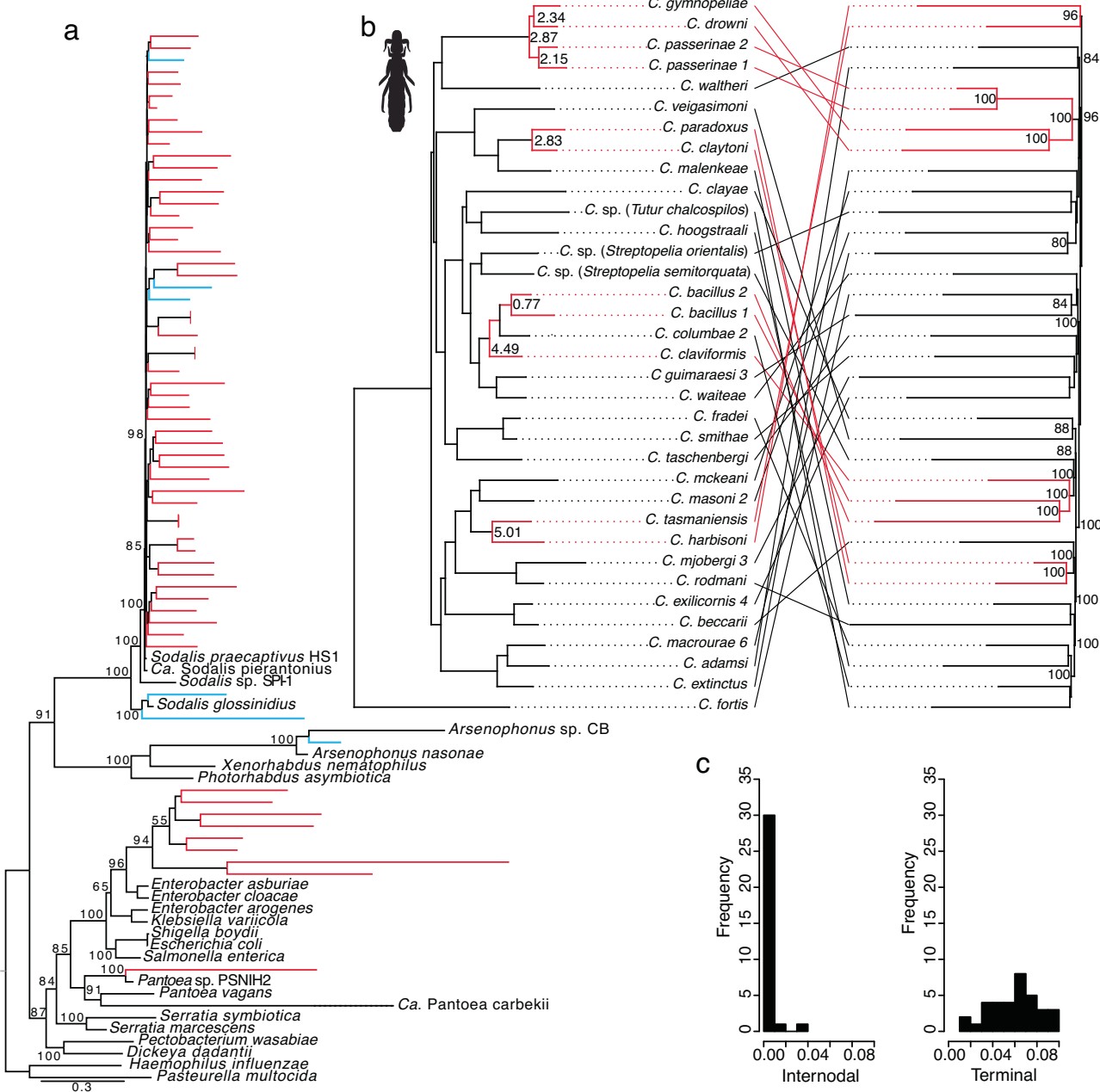

**Fig. 1 | Phylogenetic reconstructions showing relationships of louse endo-symbionts to other bacteria and comparisons to the evolutionary history of host species. a** Maximum-likelihood phylogeny of louse endosymbionts and representative Enterobacterales based on 13 single copy orthologs. Red tips indicate endosymbionts from *Columbicola* species and blue tips indicate endosymbionts from feather-feeding lice not belonging to the genus *Columbicola*. Numbers at nodes indicate bootstrap support. Branch length to the root has been shortened and some node and tip labels have been omitted to facilitate presentation. **b** Tanglegram comparing a phylogeny of *Columbicola* lice (left) with the phylogeny of their louse endosymbionts (right). Louse tree is a maximum-likelihood phylogeny based on 977 loci[28]. Numbers at nodes of louse tree indicate species divergence times in millions of years before present for predicted host-endosymbiont cospeciation events (dates ranged from 1.74-3.22 or 2.83-5.01

million years ago, depending on the phylogenetic calibrations employed; latter age range shown)[28]. Endosymbiont tree is a maximum-likelihood phylogeny of endosymbionts closely related to *S. praecaptivus* and is based on 297 universally conserved single copy orthologs. Numbers at nodes of endosymbiont tree indicate bootstrap support >80%. Red branches and connecting bars indicate louse-endosymbiont cospeciation events. Statistical comparisons of louse endosymbiont trees showed they were not more similar than would be expected by chance (ParaFit Global Fit = 0.02, *P* = 0.24, based on 999 randomizations of host-parasite associations; JANE Cost=126). **c** Histograms showing frequency of branch lengths (internodal, left and terminal, right) from the endosymbiont tree presented in panel b right-hand side. Abbreviations: *Ca.* = *Candidatus*, *C.* = *Columbicola*, sp. = species. Source data are provided as a Source Data file.

a largely convergent manner, leading to the retention of 298 universally intact protein-coding genes. Out of a total of 3932 chromosomal protein-coding genes identified in *S. praecaptivus*, 3645 had some fraction of these genes retained either in an intact (functional) form or as pseudogenes in at least one endosymbiont, strongly

supporting the notion that ancestors of these endosymbionts had gene inventories closely matching *S. praecaptivus*. In addition, few remnants of a *S. praecaptivus* plasmid sequence were detected, implying universal loss of this element in transition to symbiosis. *S. praecaptivus* also maintains two prophages within its chromosome,

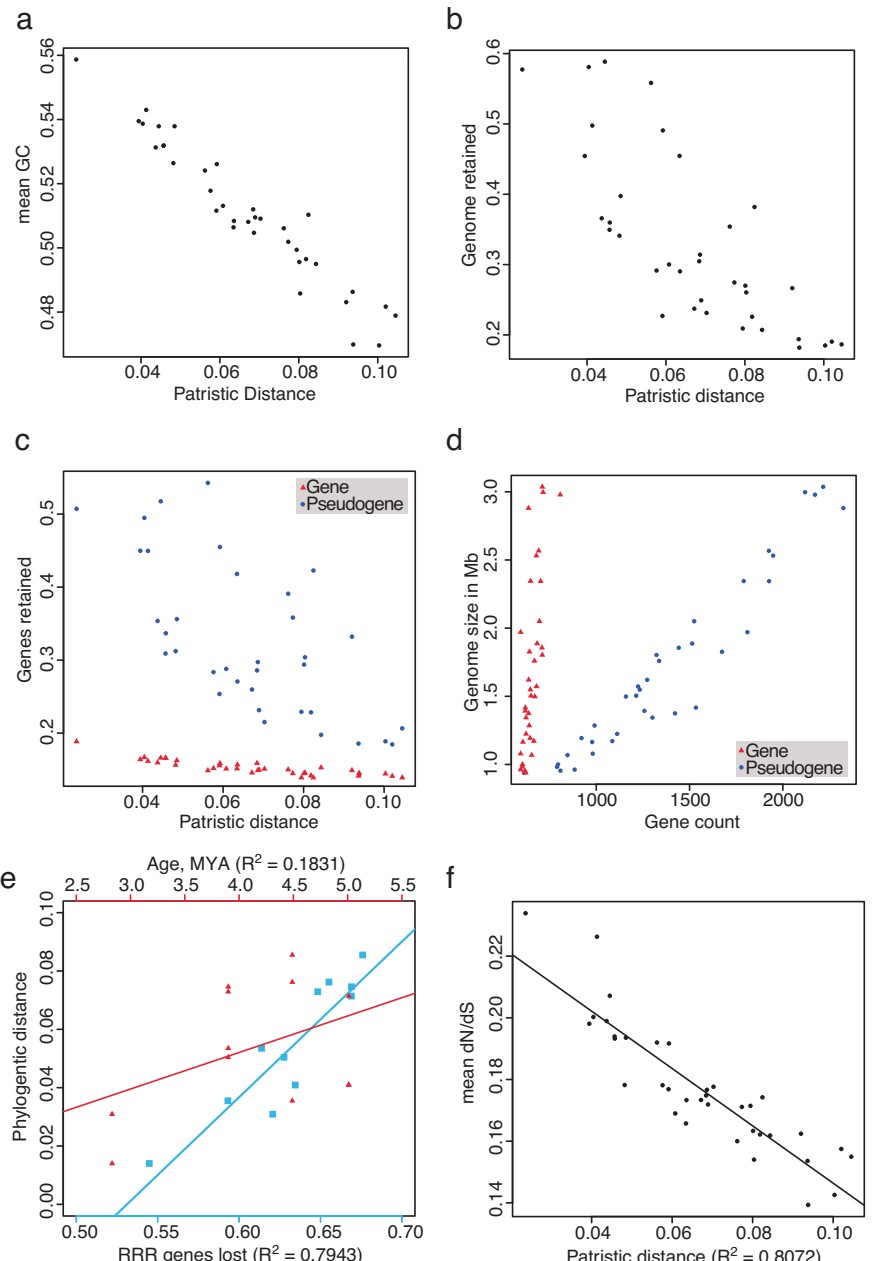

**Fig. 2 | Comparison of evolutionary distances and genomic features in endosymbionts, and comparisons of genome features. a** Mean fraction of GC bases in 297 universally conserved orthologs in endosymbiont compared to patristic distance separating the endosymbiont from *S. praecaptivus*. **b** Fraction of the genome of *S. praecaptivus* recovered in endosymbionts (total length of the aligned reads / total length of the *S. praecpativus* genome) compared to patristic distance separating the endosymbiont from *S. praecaptivus*. **c** Fraction of genes found in *S. praecaptivus* discovered in endosymbionts (either intact or as a pseudogene) compared to patristic distance separating the endosymbiont from *S. praecaptivus*. **d** Genome length compared to gene content in endosymbionts. **e** Fraction of replication, repair, and recombination genes found in *S. praecaptivus* discovered in endosymbionts derived from louse-endosymbiont cospeciation events (x-axis, red, tics at top of plot) and time since cospeciation events in millions of years (x-axis, blue, tics at base of plot), compared with phylogenetic distance (y-axis). **f** Mean dN/dS for 297 single copy orthologs universally conserved in *S. praecaptivus* and closely related louse endosymbionts compared to patristic distance separating the endosymbiont from *S. praecaptivus*. In **a**–**c** and **f** patristic distances on x-axis are defined as the distance between an endosymbiont and *S. praecaptivus* in the tree illustrated in Fig. 1b. In **e**, phylogenetic distance is defined as the patristic distance between the endosymbiont tip and the node from which all endosymbionts diverge. Abbreviations: MYA = million years ago, RRR = repair, replication, and recombination. Source data are provided as a Source Data file.

only one of which is well represented in the genomes of the louse endosymbionts. This trend fits with many observations of endosymbiont genome degeneration and base compositional bias observed in other associations[3]. Also, pseudogene abundance accounts for most of the variation in endosymbiont genome sizes (Fig. 2c, d), indicating that endosymbionts maintain similarly sized functional gene inventories, but have experienced different levels of pseudogene loss via deletions.

Variation in branch length and other genomic characteristics observed among the *Sodalis*-allied endosymbionts of lice is often inferred to be a consequence of different ages of association such that endosymbionts on the shortest branches are predicted to represent those that are most recent in origin. To evaluate this notion, we focused on endosymbionts derived from louse-endosymbiont cospeciation events with known divergence dates. However, we found that time was not significantly correlated with patristic distance in these

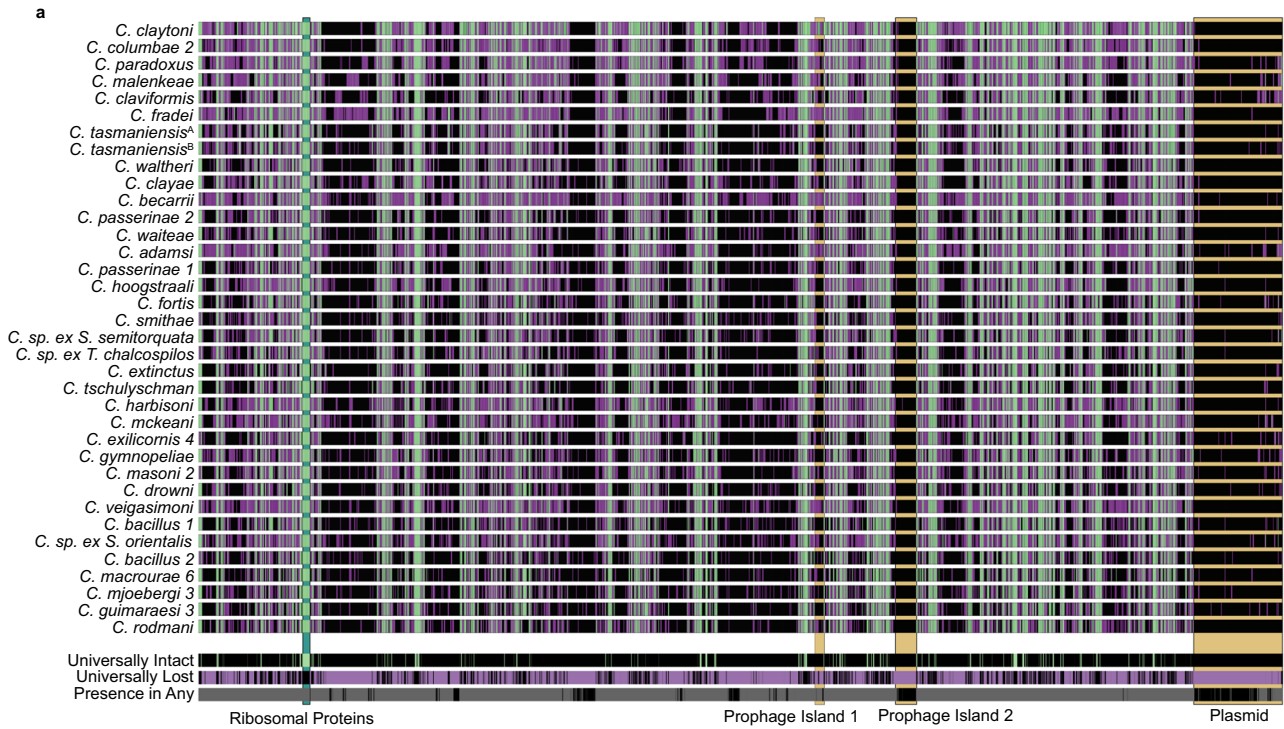

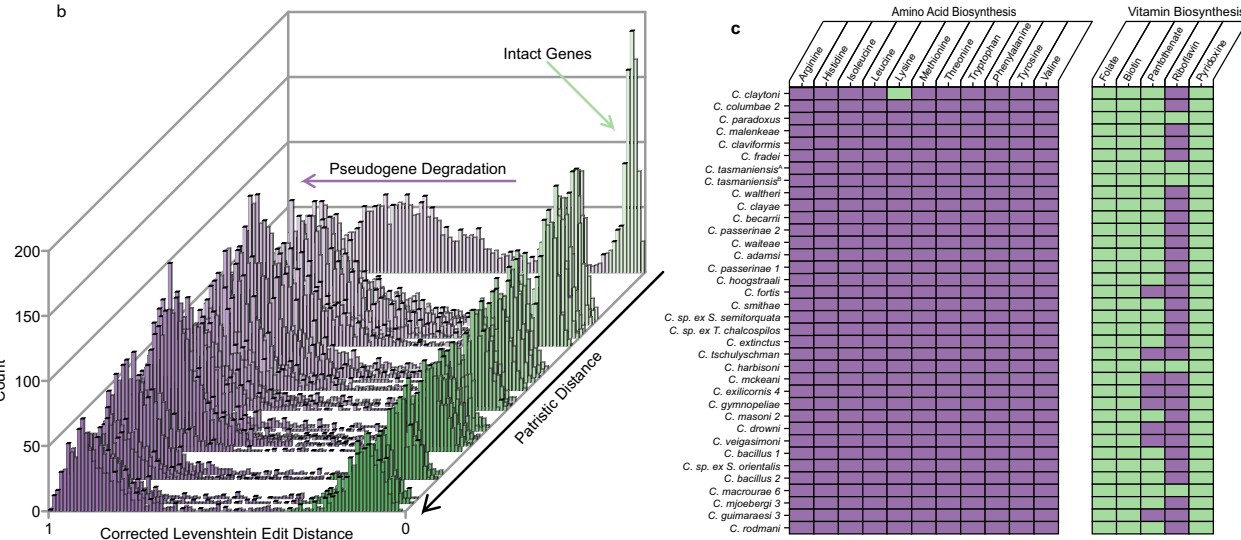

**Fig. 3 | Reduced gene inventories and functions in *Columbicola* endosymbionts.**
**a** Matrix displaying endosymbiont gene inventories mapped to *S. praecaptivus* chromosome and plasmid arranged in order of patristic distance (from top) showing intact genes in green, pseudogenes in purple and missing genes in black. Summary information, derived from all 36 individual louse endosymbionts, is presented in the lower three rows, comprising universally intact genes (top; green), genes universally lost (middle; purple) and genes that are present, at least fractionally, in any endosymbiont (gray; bottom). **b** Histogram showing the numbers of genes with given corrected Levenshtein Edit Distances (cLEDs) among the 36 endosymbionts arranged in accordance with patristic distance from *S. praecaptivus* (back to front). Genes predicted to be intact are shown in green and those predicted to be pseudogenes are shown in purple. Note that as patristic distance increases, the cLEDs of pseudogenes are observed to increase, consistent with increased levels of deletion. **c** Predicted status of amino acid and B vitamin biosynthesis pathways. Pathways describing amino acid biosynthetic enzymes are annotated as intact (green) only if genes encoding all biosynthetic steps are predicted to be intact in the respective symbiont genome. Pathways describing B-vitamin biosynthesis are annotated as intact (green) if genes encoding all biosynthetic steps are intact in the respective symbiont genome and/or present in a representative (*C. columbae*) host sequence. Note that in panels **a** and **c**, *C. tasmaniensis*[A] and *C. tasmaniensis*[B] are endosymbionts of lice obtained from closely related dove species: *Phaps chalcoptera*[A] and *P. elegans*[B]. Abbreviations: *C.* = *Columbicola*, sp. = species, ex. = isolated from. Source data are provided as a Source Data file.

examples ($R^2 = 0.1831$, $F = 2.242$, $P = 0.2652$; Fig. 2e), suggesting that other factor(s) influence the rate of mutation in these endosymbionts. Also, the daughter branches resulting from cospeciation events demonstrate substantial variation in length (e.g., *C. bacillus* 1 and *C. bacillus* 2 endosymbionts, *C. paradoxus* and *C. claytoni* endosymbionts; Fig. 1b), indicating that substitution rates can change

following cospeciation events, and therefore likely differ more broadly among symbionts in different hosts.

## Gene loss and replicational fidelity
Numerous studies show that genome degeneration is accompanied by loss of DNA recombination and repair mechanisms, leading to

increased mutation rates and compromising the processivity of replication[9,20,29,30]. Furthermore, some endosymbionts lose components of the replisome, notably the initiation protein DnaA, which coordinates the timing and topology of replication[20]. To investigate the impact of these losses in the louse endosymbionts, we identified genes classified as functioning in recombination, repair, and replication (RRR) processes in *S. praecaptivus* and then determined the fraction of these genes retained in louse endosymbionts. We found that the patristic distances derived from phylogenetic analyzes were correlated with the number of RRR genes retained in each endosymbiont ($R^2 = 0.7943$, $F = 38.6$, $P = 9.971 \times 10^{-5}$; Fig. 2e). This suggests that loss of these functions could serve to accelerate rates of substitution, potentially accounting for the observed heterogeneity in evolutionary rate. This also fits with a broader theory postulating that an organism's proteome size is a key determinant of DNA repair capability, such that organisms with reduced numbers of protein coding genes (including endosymbionts with highly reduced proteomes) can tolerate reduced replicational/repair fidelity[31]. For comparative purposes, we also counted the numbers of genes functioning in translation and found no significant correlation with patristic distance and endosymbiont genome size (data available in the data repository).

## Selection within protein coding sequences

Asexual endosymbiont populations are predicted to be subject to Muller's Ratchet, in which deleterious mutations accumulate due to frequent population bottlenecks increasing genetic drift[14,32–34], potentially leading to mutational meltdown[33]. However, endosymbiotic bacteria maintain mutualistic associations with their hosts for hundreds of millions of years[15], suggesting that the effects of Muller's Ratchet are ameliorated by the large effective population sizes of host insects[34].

To explore the effect of natural selection on genome evolution in the endosymbionts, we computed dN/dS ratios from pairwise alignments of 297 universally-retained, intact genes shared between *S. praecaptivus* and the louse endosymbionts. Generally, dN/dS provides an assessment of the relative contributions of selection and drift on DNA substitutions[35]. Strikingly, across louse endosymbionts, dN/dS is observed to be inversely correlated with patristic distance (i.e. total substitutions, $R^2 = 0.8072$, $F = 142.4$, $P = 1.055 \times 10^{-13}$; Fig. 2f), indicating that endosymbionts localized on longer branches (with smaller genomes) are subject to increased stabilizing selection. This pattern occurs despite the fact that Muller's Ratchet is predicted to lead to the loss of DNA repair and recombination systems, increasing the rate of accumulation of deleterious mutations[36]. Here, we propose two (non-mutually exclusive) explanations for the observed negative correlation between dN/dS and patristic distance. First, the transition to endosymbiotic life creates a window of opportunity for adaptive mutation to optimize gene functions in a novel environment[37]. However, once key fitness gains are obtained, selection should resume a stabilizing role, reducing the relative rate of amino acid changing (nonsynonymous) substitutions[38]. Second, a smaller genome may allow more copies of the genome to be maintained in each cell. Thus, it is possible that drift, resulting from population bottlenecks encountered during endosymbiont transmission, could be ameliorated by endosymbionts transitioning to polyploidy[39,40].

## Deterministic genome degeneration

To infer the nutritional basis of the feather louse symbiosis, we inspected the status of endosymbiont genes encoding components of biosynthetic pathways yielding amino-acids and vitamins (Fig. 3c). Feather lice feed on a diet composed of feather keratin[41], which is nutritionally complete with respect to amino acids[22]. Not surprisingly, all louse endosymbionts were found to lack amino acid biosynthesis pathways, except those components yielding intermediate metabolites (i.e. chorismate and diaminopimelate) that are used in B-vitamin

and peptidoglycan synthesis. Conversely, genes encoding components of folate and biotin biosynthesis pathways are intact in all louse endosymbionts and genes encoding components of riboflavin and pantothenate biosynthesis pathways are intact in the majority of louse endosymbionts. Together these findings suggest the primary function of the feather louse *Sodalis* endosymbionts is to produce B-vitamins to augment the protein-rich diet of their hosts (data available in figshare).

## Contingency in genome degeneration

Genome degeneration is anticipated to be a largely stochastic process resulting from deactivation of genes and deletion of sequences under relaxed selection. This degeneration could yield distinct outcomes among different endosymbionts as a consequence of stochasticity (random mutations) and functional contingency[42]. To investigate this issue, we focused on characterizing 887 genes that are fractionally retained among the louse endosymbionts; i.e. inactivated or lost in at least one lineage but retained in a functional state in others. Following exclusion of two taxa derived from recent cospeciation events, relationships between fractionally retained genes in 34 endosymbionts were classified to identify cases where endosymbionts have alternating patterns of gene retention (i.e. each endosymbiont retains gene A or gene B), herein described as "reciprocal" relationships. Direct relationships occur for functionally co-dependent genes (e.g. genes constituting operons or encoding essential components of a protein complex), such that loss of one gene makes another obsolete, enabling inactivation/loss. Conversely, reciprocal relationships, in this context, occur between genes with redundant functions, allowing loss of one or the other, but not both.

To identify direct and reciprocal relationships, binary string classification was performed by computing a fully recursive set of Hamming distances among 887 strings. Data visualization (Fig. 4) reveals an extensive network of direct and reciprocal relationships among those fractionally retained genes. As expected, direct relationships are observed between genes performing common or code-pendent functions, validating this approach. One interesting exception involves *mutL*, which functions together with *mutH* and *mutS* in methyl-directed mismatch repair[43] (MMR). Unexpectedly, *mutL* was classified as inactivated in endosymbiont lineages that retain intact *mutH* and *mutS* genes (Fig. S5). However, further inspection revealed that the classification was inaccurate because *mutL* sequences maintain in-frame deletions in a region of the protein corresponding to a linker residing between structural domains. Interestingly, in *Escherichia coli*, truncation of this linker reduces the ability of MutHSL to bypass "roadblocks" (DNA-binding proteins or complexes including RNAP) that interfere with MMR[44]. These results demonstrate that our analytical methods can reveal novel and detailed insight into protein functions and molecular pathways in cells.

Many genes were also identified that demonstrate significant reciprocal relationships based on analysis of Hamming distances (data available in figshare) and Proximus clustering analysis (data available in figshare), which groups genes with retention patterns sharing significant direct and inverse similarity into sets in accordance with Hamming radius. Strikingly, genes demonstrating reciprocal patterns of retention are often found to perform analogous functions, indicating that the process of genome degeneration in the louse endosymbionts has promoted the depletion of functional redundancy. Among many examples that arise from our analyzes, we highlight two clear and insightful examples of this phenomenon, involving (i) genes encoding electron transport chain components (Fig. 4) and (ii) genes encoding DNA repair and recombination enzymes (Fig. 5).

One highly-significant and prominent feature of the electron transport chain contingency centers upon reciprocal retention of the genes encoding the Nuo and Ndh NADH dehydrogenases. The *nuo* genes encode a 14-subunit, proton-pumping NADH:ubiquinone oxidoreductase, which is the first respiratory chain complex in bacteria

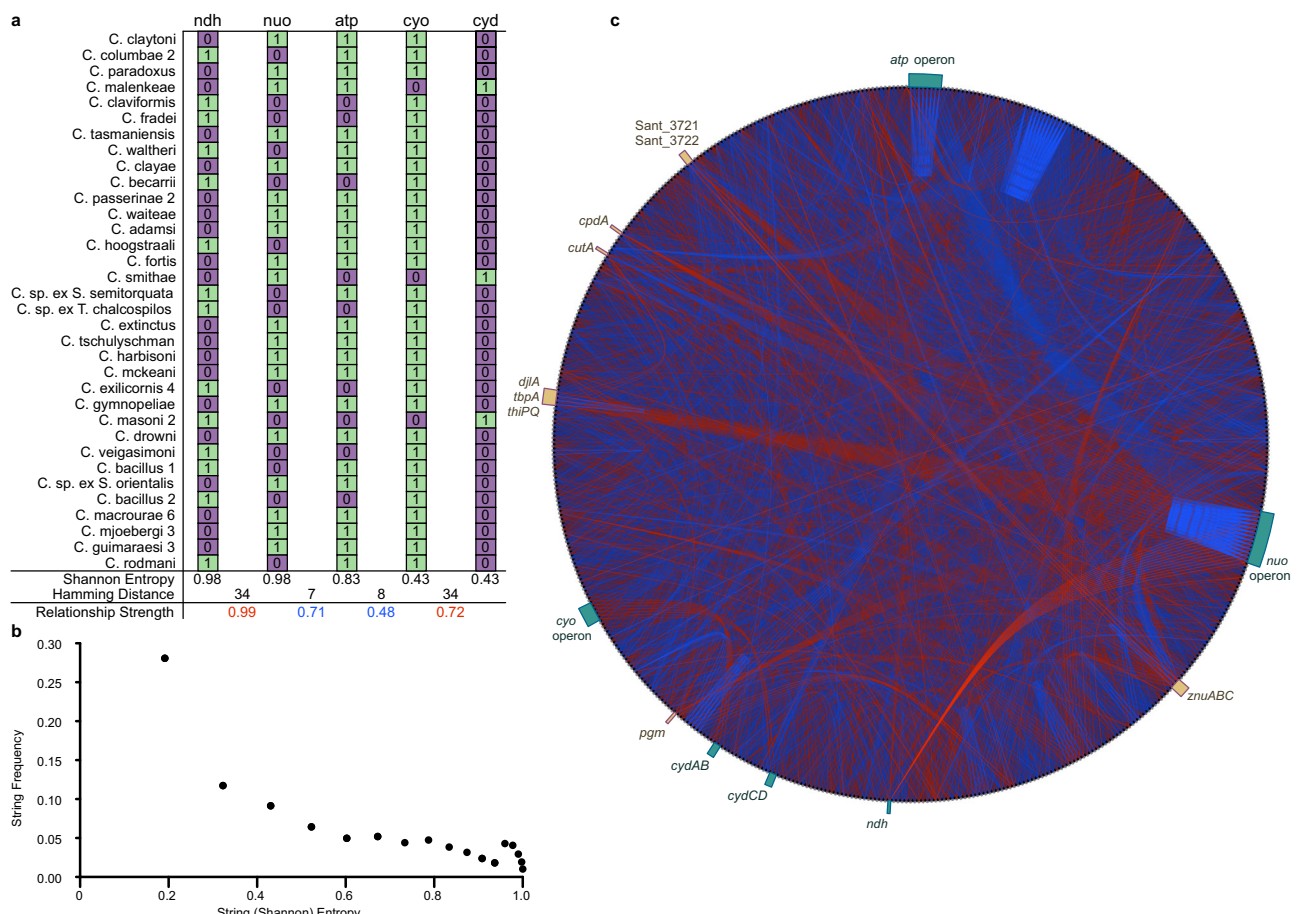

**Fig. 4 | Identification of genes with direct and reciprocal retention patterns among the louse endosymbionts. a** Matrix depicting functional status of genes encoding respiratory chain components (1/green = functional, 0/purple = pseudogene or absent) in each louse endosymbiont. Binary strings derived from each gene are analyzed to determine string (Shannon) entropy (E) and Hamming distances (H) between strings leading to derivation of relationship strength as

$$0.5E + \frac{0.5}{17}\sqrt{(H-17)^2}$$ (see Supplementary Materials). **b** Plot depicting frequencies of strings with different string (Shannon) entropies, showing that low entropy strings are overrepresented in the dataset. **c** Circle plot showing relationships between genes that are fractionally retained among the louse endosymbionts, highlighting

co-retention (blue) and reciprocal retention (red). Only genes whose binary strings have entropies ≥ 0.43 (corresponding to at least three cases of retention or loss) and relationships with Hamming distances ≠ 17 are rendered. Color intensity reflects relationship strength as defined in (**a**) and determines the order of rendering in the plot. Genes functioning in the respiratory chain are highlighted in teal whereas genes with other functions are highlighted in tan. Regarding the latter, several genes predicted to function as ATP-powered transporters (*znuABC*, Sant_3721, Sant_3722, *tbpA* and *thiPQ*), along with genes encoding other proteins known to influence energy homeostasis (*cutA*, *cpdA*, *pgm* and *djlA*). Abbreviations: *C.* = *Columbicola*, sp. = species, ex. = isolated from. Source data are provided as a Source Data file.

and mitochondria[45]. Loss of any of the *nuo* genes deactivates the complex; therefore, genes encoding Nuo subunits are either universally retained in an intact/functional form or universally inactivated/lost. In endosymbionts in which Nuo is inactivated, we find invariably that an alternative NADH dehydrogenase, Ndh, is maintained. Interestingly, Ndh is a non-proton pumping NADH dehydrogenase that also delivers electrons to the respiratory chain, but fails to contribute to an electrochemical (proton) gradient[46]. Almost all free-living bacteria maintain both Nuo and Ndh, but their differential roles are not fully understood. In *Pseudomonas aeruginosa* it was demonstrated that loss of Nuo yields a significant anaerobic growth defect[47]. In addition, Ndh might facilitate increased carbon flux into metabolic pathways under conditions in which Nuo activity is inhibited by a high proton-motive force[46]. We also observe Nuo retention to be strongly linked to retention of genes encoding the $F_1F_0$-type ATP synthase complex, another important respiratory chain component that generates ATP using the proton-motive force procured in large part by Nuo[48]. Furthermore, our analysis reveals loss of genes encoding the ATP synthase complex to be linked to retention of *cyd* genes encoding the cytochrome bd oxidase complex and to loss of *cyo* genes encoding the cytochrome bo oxidase complex. Consistent with this

finding, cytochrome bo translocates $2H^+/e^-$, whereas cytochrome bd translocates only $1H^+/e^-$ [49]. In addition, several other genes have retention profiles sharing significant direct or reciprocal similarity to Nuo. These include: *pgm* (α-phosphoglucomutase), which plays a significant role in directing carbon flux[50]; numerous transporters driven either by ATP or the proton-motive force; and *cpdA*, a cyclic AMP phosphodiesterase that regulates expression of the cytochrome bd and cbb3 oxidases that function downstream of Nuo or Ndh in the respiratory chain. In addition, the retention pattern of the $F_1F_0$ ATP synthase genes is largely reciprocal to that of the gene encoding the ATP-dependent primosomal helicase PriA and the ATP-utilizing sensory kinase (PhoQ), which is known to participate in antimicrobial peptide resistance in *Sodalis* endosymbionts[51]. Taken together, these results indicate that mutations inactivating respiratory chain components have wide-ranging contingent effects on the retention of other genes involved in energy generation, establishment of the proton motive force, central metabolism, transport, replication, and transcriptional regulation.

In the second key category of contingencies, the loss of DNA repair and recombination genes in the louse endosymbionts is both varied and extensive, but we identified multiple contingencies. For

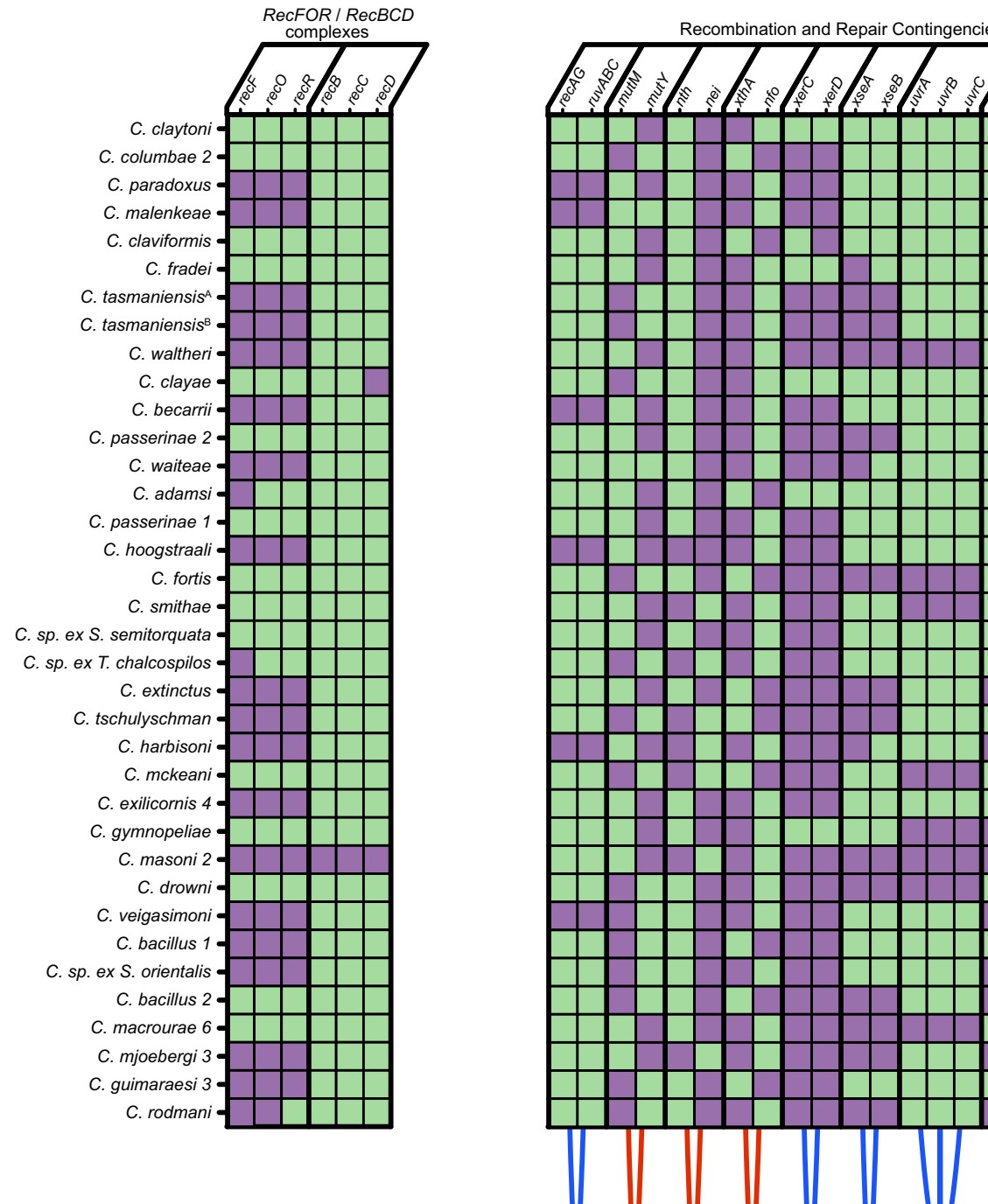

**Fig. 5 | Co-retention and contingent loss of genes in endosymbiont genomes involved in DNA repair and recombination.** Genes predicted to be intact are shown in green, and genes predicted to be inactive or lost are shown in purple. Blue arcs at the bottom indicate genes that are co-retained, while red arcs indicate genes that are reciprocally retained. Hamming distances are listed below the arcs, indicating the strength of direct or reciprocal similarity with zero representing a perfect match and 34 representing perfect reciprocality. *C. tasmaniensis*[A] and *C. tasmaniensis*[B] are endosymbionts in lice collected from closely related species of doves: *Phaps chalcoptera*[A] and *P. elegans*[B]. Abbreviations: *C. = Columbicola*, sp. = species, ex. = isolated from. Source data are provided as a Source Data file.

example, among recombination functions, our results demonstrate that the RecFOR pathway is lost much more frequently than RecBCD. Surprisingly, loss of the RuvABC resolvasome is always accompanied by loss of RecA and RecG. Thus, many of the louse endosymbionts have lost a key mechanism catalyzing replication fork reversal, which allows replication to resume when the replisome becomes dissociated from DNA due to lesions or proteins bound to DNA, such as

transcription complexes[52]. In addition, several DNA repair gene sets with similar functions show near-perfect reciprocal contingencies, indicating that for a given particular mode of repair, elimination of functional redundancy is selectively viable. For example, MutM and MutY independently protect against GC to TA transversions resulting from incorporation of oxidatively damaged guanine, 7,8-dihydro-8-oxoguanine (8-oxoG). Loss of these genes exposes bacteria to an

underlying AT-biased mutational profile[29]. All except two of the louse endosymbionts lack either MutY or MutM, resulting in a mutator phenotype anticipated to increase relative rates of those DNA substitutions, leading to AT richness, as often observed in endosymbionts[3]. However, none lack both MutY and MutM, which results in a mutator phenotype that is 25 to 75 times more potent[53]. Similarly, the louse endosymbionts show near perfect reciprocal retention of the Nth and Nei base excision repair endonucleases that target lesions originating from oxidized pyrimidines. *E. coli* mutants lacking Nth or Nei have a mutator phenotype that is 20 times more potent in double mutants[54]. Finally, the louse endosymbionts demonstrate perfect reciprocal loss of genes encoding the XthA and Nfo exo- and endonucleases, respectively, that repair apurinic/apyrimidinic sites resulting from oxidative damage[55]. Again, in *E. coli*, single mutants show elevated rates of mutation, while loss of both repair genes has a compounding effect[56].

Taken together, our results indicate that the elimination of functional redundancy is a consistent outcome of the process of genome degeneration in the louse endosymbionts. Indeed, our results mirror those obtained from broader comparative genomic analyzes[57] in addition to a theoretical study that simulated genome evolution in the confines of a minimal metabolic network[42]. Importantly, loss of functional redundancy is predicted to take place when organisms transition to a monophasic lifestyle. In other words, the maintenance of functional redundancy is predicted to be advantageous under conditions in which an organism persists in a dynamic and uncertain environment that mandates "robustness", which is defined as the ability to maintain key cellular functions in the face of physiological or environmental perturbations[58].

## Discussion

Our work shows that the *Sodalis*-allied endosymbionts of feather lice evolved repeatedly and independently in closely related hosts with identical lifestyles, likely providing B-vitamins absent in the hosts' specialized diet of feather keratin. Not surprisingly, this is underscored by retention of genes encoding B-vitamin biosynthesis pathways in all louse endosymbionts analyzed in our study. Numerous studies reveal that bacteria undergo extensive genome degeneration resulting from the lifestyle transition to obligate host association[9,20]. This process leads to functional reduction, such that only those genes having adaptive value in the symbiosis are retained. While the process is gradual in nature, it is predicted to be accelerated by frequent population bottlenecks imposed during vertical endosymbiont transmission that reduce selection efficiency[9].

In spite of these predictions, our study showed that louse endosymbionts maintain a striking level of protein family diversity, attributable to selection pressure facilitating loss of redundant gene functions. Thus, the trajectory of gene loss in the louse endosymbionts can be likened to a decision tree with nodes arising initially from chance events (stochastic null mutations), providing opportunities for subsequent losses/nodes that are influenced by functional contingencies. More broadly these findings support the notion that genomic and functional degeneration may be adaptive in symbiosis[36], potentially providing a source of beneficial mutations that compensate for deleterious mutations predicted to arise at increased frequencies in endosymbionts due to Muller's Ratchet[59].

In summary, this study leveraged a highly repetitious evolutionary process to gain detailed insight into the trajectory of genome degeneration in symbiosis. Our findings show that the forces shaping genome degeneration are both stochastic and deterministic. These forces yield distinct phenotypic outcomes that differ largely as a consequence of gene losses influenced by historical contingency. More broadly, our investigation of replays of the "tape of symbiosis" in feather-feeding lice aligns with the notion that evolutionary processes are subject to diverse outcomes as a consequence of historical

contingency[24,25]. Notably, in the case of the louse endosymbionts, contingency influenced the retention of codependent functions, leading to distinct endosymbiont gene inventories featuring diversity in their metabolic underpinnings that nonetheless perform the same mutualistic functions in closely-related hosts.

## Methods

### Taxon selection and identification of endosymbionts

Several studies[5,21,60,61] have characterized endosymbionts from feather feeding lice in the genus *Columbicola* using light microscopy, analysis of 16S rDNA sequences, and fluorescent in situ hybridization imaging. These studies reveal that *Columbicola* lice host gamma-proteobacterial endosymbionts from the genus *Sodalis*, or two other genera in the Enterobacterales. These endosymbionts are difficult to isolate from louse tissues and are present in relatively small numbers; however, several studies[13,22,62-64] show that when total genomic DNA is sequenced from whole lice, endosymbiont genomes are represented within the genomic DNA providing a basis for our approach.

We used existing whole genome shotgun, Illumina sequence libraries prepared from whole lice to examine the bacterial associates in 61 different samples of *Columbicola*, collected from avian species worldwide (see Boyd et al.[27] for complete sequencing methods), available on the NCBI short read archive (https://www.ncbi.nlm.nih.gov/sra; SRP069898; Table S1). The archive included 47 samples from 45 described species of *Columbicola* (two *Columbicola* species were each sampled from two different avian host species) and 14 potentially undescribed *Columbicola* species. Potentially undescribed species were either identified in the data set as a species (sp.) or with a number following the species names. This latter class represents cryptic species identified previously[27], based on P-distance calculated from the mitochondrial Cytochrome c oxidase I gene. Seven additional feather louse species from other genera were also included to provide a broader phylogenetic context in which to interpret results from *Columbicola*, and to identify additional louse species that might be host to *Sodalis*-endosymbionts (Table S1).

We used a targeted locus assembly method, aTRAM[65,66], which provided up to 13 different single-copy orthologs from each library and representative Enterobacterales, which were used to construct a phylogenetic tree using RAxML (Fig. 1a, S1)[67]. From this tree, we identified the diversity of louse endosymbionts from four different clades of Enterobacterales. We selected louse species containing endosymbionts that were closely related to *S. praecaptivus* strain HS1, a close free-living relative of the *Sodalis*-allied insect endosymbionts[68], for whole genome reconstruction using novel read-mapping and alignment procedures.

We then validated our phylogenetics results using 241 single-copy orthologs from the most complete genome assemblies (genome assembly, annotation, and ortholog selection described in Supplementary Materials). Individual orthogroups were then aligned as translated amino acid sequences using MUSCLE (v.3.8.31)[69], and converted back to nucleotides. Third positions were excluded due to variations in base composition between lineages. RAxML (v.8.2.12) was used to determine shape and rate parameters for the first and second codon positions in each gene. Principal components analysis was used to find the optimal data partitions based on alpha and rate parameters, finding that eight partitions were optimal. This result was confirmed using K-means clustering by calculating the between sum of squares over the between sum of squares for all possible K-means clusters, again finding that eight partitions were optimal. Finally, each codon alignment was grouped into partitions using k-means clustering (gene alignments and partition file are provided in the figshare data repository; all statistical analyzes were conducted using R v.4.0.3)[70]. A maximum-likelihood tree was generated using RAxML based on a partitioned analysis of a concatenation of all sequence data under a GTR-G model of sequence evolution. Support for phylogenetic

relationships was determined by the percentage of 100 bootstrap replicate trees that supported each node. We then replicated the analysis using best-fit time-reversible models of sequence evolution using IQ-TREE (v.1.6.12)[71–74]. Additionally, aligned amino acids were also used to create a supermatrix, with alignment merging and best-fit models being identified by IQ-TREE. Support for trees inferred using IQ-TREE was assessed using 1000 ultrafast bootstrap replicates.

### Evolutionary history of *Sodalis* isolated from *Columbicola*

Comparative genomics facilitated identification of 297 single-copy orthologs retained among the *Sodalis*-allied *Columbicola* endosymbionts, and the *S. praecaptivus* progenitor[68]. Of these, 297 were used in phylogenetic analyzes. One gene (*prfB*) was excluded, as it is known to have a programed translational frameshift in many bacteria, but this has not been established in *Sodalis*. Each gene was translated to an amino acid sequence and aligned using MUSCLE, and back-translated as an aligned sequence. The mean and variance for the frequency of GC bases was calculated by species for each codon position (Fig. S6). Third positions were excluded due to variation in base composition between lineages. RAxML was used to determine shape and rate parameters for first and second codon positions in each gene. Principal components analysis was used to find the optimal data partitions based on alpha and rate parameters, finding that eight partitions was optimal. This result was confirmed using K-means clustering, by calculating the between sum of squares over the between sum of squares for all possible K-means clusters, again finding that eight partitions was optimal. Finally, each codon alignment was grouped into partitions using k-means clustering (gene alignments and partition file are provided in the figshare data repository; all statistical analyzes were conducted using R v.4.0.3). A maximum-likelihood tree was generated using RAxML based on a partitioned analysis of a concatenation of all sequence data under a GTR-G model of sequence evolution. Support for phylogenetic relationships was determined by the percentage of 100 bootstrap replicate trees that supported each node. Bootstrap replicates with partitions were generated using RAxML and ML trees were generated in the same manner as the original matrix. All phylogenetic trees were visualized using Fig-Tree (v.1.4.3).

### Effect of host on endosymbiont diversification

We compared the host phylogeny and the *Sodalis* phylogeny to identify cospeciation events and assess the level of congruence between host and endosymbiont trees[75,76]. For all comparisons of the host and endosymbiont trees, we used the *Sodalis* tree based on 297 single copy loci (not the original tree based on 13 loci) and a host tree described in Boyd et al.[27], based on the same original sequence data, and constructed from 977 single copy loci. Endosymbionts from conspecific lice could not be used for an analysis of codiversification, therefore, one sample of *C. tasmaniensis* was omitted.

To test for overall congruence between louse and endosymbiont phylogenies, we used the distance-based method PARAFIT[75], available in the R package ape[77]. PARAFIT assumes two phylogenies are randomly associated and tests for the contribution of each association (link) to the overall (global) congruence. For all PARAFIT analyzes, we converted phylogenetic trees to patristic distance matrices in the R package ape (cophenetic.phylo), and ran PARAFIT for 999 iterations. We used the "Cailliez" correction for negative eigenvalues, and tested for the contribution of individual links with the ParaFitLink1 and Para-FitLink2 tests (data files are provided in the figshare data repository).

We also tested for cospeciation events using the event-based method JANE (V.4)[76] with the same trees. This method seeks to optimally reconstruct evolutionary events (cospeciation, host switching, duplication, and sorting events) between phylogenies of two groups of interacting organisms by minimizing the overall cost of events (cost parameters determined a priori). Because it is extremely unlikely that

host-switching is a plausible scenario in the louse-endosymbiont system, we restricted JANE from considering any host-switching events by increasing the host-switching cost parameter to 15. The remaining cost parameters were kept at default values (0 for cospeciation events, 1 for duplication, 1 for losses, and 1 for failure to diverge). The louse-endosymbiont system also likely involved multiple replacements from a free-living bacteria ancestor, as demonstrated by our phylogenetic and cophylogenetic results. Any loss or duplication events could be interpreted as replacement, but because JANE does not explicitly model replacements, these should be interpreted cautiously. For our JANE analyzes, we set the Genetic Algorithm (GA) parameters to generations = 500 and population size = 1,000. We tested whether the total JANE event reconstruction cost was significantly less than random by randomizing the tip associations 100 times.

### Evolutionary distances, time, and genomic features

We calculated the patristic distance between each *Sodalis* endosymbiont and the closely related and free-living *S. praecaptivus*, which provided a consistent phylogenetic comparison of genome composition. Genome size, gene content, and base composition were compared to patristic distances. Patristic distances were calculated using R package ape (cophenetic.phylo, patristic) using the tree resulting from the 297 gene supermatrix. Ratios of non-synonymous to synonymous substitutions (dN/dS) for 297 universally conserved genes were calculated between *S. praecaptivus* and the endosymbiont genomes using the R package ape (function dnds), which computes the dN/dS following the method described by Li[78]. Statistical analysis was conducted using R (v.4.0.3).

Twelve endosymbionts of *Columbicola* were identified as being derived from louse-endosymbiont cospeciation events using JANE. The age of cospeciation events was inferred from louse speciation events, with ages for these events being obtained from two different phylogenetic analyzes of lice (results of both analyzes described by Boyd et al.[28]. in their figures S5 and S8). This approach provided us with two sets of dates for each event, with the older dates being used to test for a relationship between time and the phylogenetic distance. Distances were calculated as the patristic distance between the free-living relative *S. praecaptivus* and the endosymbiont, subtracting 2*(tip length of *S. praecaptivus*), yielding the distance between the tip of the tree and the base of tree. We then calculated the total distance between the tip and the oldest cospeciation event for each of the 12 instances, in millions of years. In 11 of 12 cases we found that the phylogenetic distance following a cospeciation event represented >80% of the overall patristic distance, and in 9 of 12 instances, that distance represented >90% of the overall patristic distance (Fig. S7), facilitating statistical comparisons of phylogenetic distances and time in millions of years.

### Genome alignments

We developed a novel pipeline to align endosymbiont sequence reads to the whole genome sequence of *S. praecaptivus*. This is based on the notion that the louse symbionts are derived from a free-living ancestor (progenitor) that is a close relative of *S. praecaptivus*, as evidenced by our phylogenetic analyzes. This also assumes that endosymbionts undergo degenerative evolution, such that their gene inventories are subsets of *S. praecaptivus*[51]. Thus, our approach sought to determine the protein-coding gene content of the louse endosymbionts by aligning their sequences to the *S. praecaptivus* reference. We used alignment parameters that were optimized manually to facilitate a level of sequence mismatching and gap incorporation necessary to align >99% of the reads from all endosymbionts with uniform read depth (data files are provided in the figshare data repository).

Sequence reads from DNA libraries prepared from whole lice were aligned to a concatenated sequence comprising the *S. praecaptivus* chromosome (GenBank CP006569.1) and plasmid (GenBank

CP006570.1), using a multi-step pipeline designed to maximize the efficiency and accuracy of alignment of each endosymbiont genome. In step 1, the fastq files from whole lice libraries were quality and adapter-trimmed using Trimmomatic[79] with a minimum length filter = MINLEN:50 and quality filter = MAXINFO:50:0.2. In step 2, the length of short reads was extended by merging overlapping paired-end reads with FLASH[80]. Extended and non-overlapping read-pairs were allocated into multi-fasta files containing 500,000 reads each. In step 3, the multi-fasta files were aligned to the concatenated *S. praecaptivus* chromosome/plasmid query sequence using cross match, a banded Smith-Waterman pairwise comparison algorithm (http://www.phrap.org/phredphrap/general.html). Matches were filtered using the default score matrix and parameters for gap penalties (gap_init: −4, gap_ext: −3, ins_gap_ext: −3, del_gap_ext: −3), minimum match score (14), and minimum alignment score (30). The numbers of starting reads and filtered reads for each whole louse library are available in the data repository. Filtered reads were then binned and realigned to the *S. praecaptivus* concatenated chromosome and plasmid sequence using the Geneious mapper (v.8.1.8). General parameters comprised "fine tuning" set to "move many gaps", no read trimming, minimum mapping quality turned off, reads with best matching in multiple places mapped randomly to one of those places, up to 15 gaps of up to 1000 bp allowed per read, minimum overlap and minimum overlap identity turned off, the word length and index word length set to 13, words repeated more than 20 times were ignored, reads were permitted to have up to 25% mismatches, the maximum ambiguity of word matches was set to 4, the option to "accurately map reads with errors to repeat regions" was active, and "search more thoroughly for poor matching reads" was inactive.

### Genome sequences annotation
Endosymbiont consensus sequences were compared on a gene-by-gene basis to their *S. praecaptivus* homologs and determined to be intact, pseudogenized, or missing based on an Expectation-Maximization (E-M) analysis of their normalized Levenshtein Edit Distances (LEDs), providing a consistent and accurate framework for annotation.

Louse endosymbiont sequences that aligned to *S. praecaptivus* were exported from Geneious and annotated using a custom pipeline to identify intact genes, pseudogenes and missing genes. This identification was facilitated by comparing the sequences from all potential coding sequences in each louse endosymbiont to the intact open reading frames present in *S. praecaptivus*, using a custom Processing script (LAnner). The LAnner script first identifies all possible open reading frames (ORFs) from each putative genic sequence, using canonical and alternative bacterial start codons occurring between ± 25 bases of the start and stop codons present in the respective *S. praecaptivus* ortholog and then computes the LEDs between the louse endosymbiont and the *S. praecaptivus* sequences. The candidate louse endosymbiont ORF with the lowest LED is then considered to be the representative of the gene that is most likely to be functional. LED values from the highest likelihood candidates, encompassing all putative ORFs, are then normalized in accordance with gene size, yielding size normalized LEDs (snLEDs). Resulting snLEDs are then subjected to E-M analysis to estimate maximum a posteriori (MAP) parameters of a statistical model that anticipates a mixture of univariate normal distributions, encompassing intact genes and pseudogenes. E-M analyzes were conducted in R using "normalmixEM" in the package mixtools[81], using starting parameter means of zero and one and standard deviations of one and one, to encompass intact genes and pseudogenes, respectively. Following analysis, the MAP parameters of the distributions were used to estimate snLED values that represent cutoffs between the E-M distributions of intact genes and pseudogenes for each endosymbiont, defined as three standard deviations above the mean of the intact gene distribution. Each louse endosymbiont candidate gene was then identified as intact or pseudogenized based on the value of snLEDs relative to the established cutoffs.

### Data visualization
An interactive visual tool (GViewer) was developed in the processing integrated development environment to facilitate visualization of the status of all candidate ORFs in the louse symbionts relative to *S. praecaptivus* using the output from the LAnner/E-M analysis. This tool facilitates pan-genomic comparison of the louse symbiont genomes, identifying genes that are, for example, universally retained in an intact form, or universally degenerate or absent among the endosymbionts.

### Analysis of MutL Homologs
For analysis of the *mutL* homologs from *S. praecaptivus* and louse endosymbionts, the putative *mutL* sequences were aligned in Geneious using MUSCLE[69] with a maximum of 15 iterations and up to 10,000 MB of memory (all other parameters used default values). The predicted protein structure of the *S. praecaptivus* MutL homolog was obtained from the AlphaFold Protein Structure Database[82,83] and the predicted protein structure of the MutL homolog of the endosymbiont of *C. claytoni* was generated via AlphaFold[82] using default parameters. Structures were then compared both by visual inspection and by inspection of the predicted aligned error graphs generated by AlphaFold.

### Host-complementation in B-vitamin metabolism
Comparison of endosymbiont genomes showed that genes underlying B-vitamin metabolism were generally conserved. In some insect-endosymbiont associations, endosymbionts may lose a gene(s) involved in essential metabolism, making it appear that a pathway has been rendered nonfunctional by the loss, yet the missing gene might be complemented by activity of a host enzyme[19,84]. In order to investigate B-vitamin metabolism in this study, we sought to understand the contributions of both the endosymbiont and host. We first inspected the genome of a known louse endosymbiont, *Candidatus* Riesia pediculicola from the human head louse (*Pediculus humanus*), to develop an approximate minimal gene set required for B-vitamin biosynthesis and potential complementation by the host towards missing gene functions. Both the louse and its endosymbiont have reduced genomes (108 Mb and 0.58 Mb, respectively) and the endosymbiont has been shown to supply its louse host with B-vitamins[85]. By examining the genome of *Ca*. Riesia (genome T01218) and its host, *P. humanus* (genome T01223), we identified genes essential to B-vitamin biosynthesis and potentially complementary metabolic functions using KEGG[86–88] (https://www.genome.jp/kegg/kegg1.html) by identifying K numbers associated with genes involved in B-vitamin metabolism. The K numbers were then used to find homologous functions in *S. praecaptivus* (genome T03004), that were subsequently used to identify orthologs in the feather louse endosymbionts. Any gene in *P. humanus* that was predicted to complement a missing gene in *Ca*. Riesia was then subjected to identification in the *C. columbae* genome (queries conducted on InsectBase 2.0; http://v2.insect-genome.com; using K numbers obtained from KEGG[89].

### Comparative analysis of gene retention
Since recently diverged endosymbionts and endosymbionts from conspecific lice cannot be assumed to have evolved into endosymbiosis with lice independently, two samples were excluded from comparative analyzes. The first of these was one of two samples representing the same host and endosymbiont species (sampled from *C. tasmaniensis*) and the second, endosymbionts and host that cospeciated recently (sampled from cryptic host species *C. passerinae* 1 and *C. passerinae* 2).

To facilitate comparative analysis, the gene inventories of the louse endosymbiont orthologs of all *S. praecaptivus* protein coding genes were classified as universally retained, universally lost, or fractionally retained. The category of genes classified as fractionally retained was then subject to in-depth analysis to understand the nature of functional diversity among the louse endosymbiont. Such diversity is predicted to arise as a consequence of stochasticity and contingency in the process of genome degeneration, yielding endosymbiont genotypes that perform the same core functions for their hosts, but have distinct gene inventories. To investigate the nature of these distinctions, the predicted functional status of the fractionally retained genes was encoded in a binary matrix, encompassing 887 genes in each of 34 louse endosymbiont genomes. Preliminary evaluation of the data revealed a disproportionately large number of genes in the fractionally retained category that are retained or lost in only a small number of endosymbionts and could, therefore represent type I or II errors resulting from sequencing, alignment or E-M-based annotation procedures or the fact that null mutations are a lagging indicator of the relaxation of selection on gene function. We, therefore, elected to focus on analyzing relationships between genes with binary (Shannon) string entropies ≥ 0.43 (computed using ShanString; software available in GitHub), corresponding to at least 3/34 endosymbionts having a unique functional status for a given gene. In order to compare the gene inventories, Hamming distances were computed from string pattern vectors in an exhaustive pairwise manner using HammString (software available in GitHub). In this comparison, genes demonstrating identical patterns of retention have Hamming distances of 0, whereas those demonstrating perfectly reciprocal patterns of retention have Hamming distances of 34. The resulting dataset was then visualized in an exhaustive circle plot highlighting significant identical and reciprocal retention patterns in accordance with Hamming distances and string entropy. In addition to exhaustive visualization, the binary data matrix was analyzed using the Proximus algorithm in R (in package "cba"), which performs efficient compression of high dimensionality datasets into representative patterns to facilitate matching[90], which in this case leads to the identification of genes that are either co-retained or reciprocally-retained.

## Data availability

The raw sequence data are available from the Short Read Archive and are organized under the BioProject PRJNA296666 and metagenomic assemblies have been deposited in Whole Genome Sequences database in association with the same BioProject. BioSample, Short Read Archive, and Whole Genome Sequence accession numbers are provided in Supplementary Data 1. Data products, including phylogenetic data (gene alignments, supermatrices, tree files, partition files, and associated data), genomic data (three metagenome assemblies, assemblies resulting from the novel pipeline, and BAM files), string comparison data, and host-endosymbiont metabolic complementation data have been deposited in Figshare [https://doi.org/10.6084/m9.figshare.21830067]. Louse phylogeny and divergence time estimates were obtained from Figshare [https://doi.org/10.6084/m9.figshare.17108207] and Dryad Digital Repository [https://doi.org/10.5061/dryad.4812p]. Sample collection data, phylogenetic visualizations, protein images, genome comparisons data generated in this study are provided in the supplementary information. The genome of *Sodalis praecaptivus* (NCBI RefSeq GCF_000517425.1) was obtained from NCBI. Source data are provided as Source Data files. Source data are provided with this paper.

## Code availability

Newly generated code and associated data used to identify reciprocal and contingent gene loss are available from GitHub [https://doi.org/10.5281/zenodo.10946531].

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

## Acknowledgements

We thank Jon Seger, Fred Adler, and Adam Clayton of the University of Utah, School of Biological Sciences, Julie M. Allen of Virginia Tech, Department of Biological Sciences, and Andrew D. Sweet of Arkansas State University for feedback and advice on analysis. This work was supported by the National Science Foundation, DEB1926738 (C.D., S.E.B., and D.H.C), DEB1926919, DEB1925487, DEB13426045, and DEB1239788 (K.P.J), and Virginia Commonwealth University, Life Sciences (B.M.B). High Performance Research Computing (HPRC) Core Facilities at Virginia Commonwealth University (http://chipc.vcu.edu) were used for conducting the research reported in this work.

## Author contributions

Origination: B.M.B., K.P.J, and C.D.; data acquisitions, analysis and interpretation: B.M.B., I.J., R.B.W., S.E.B., D.H.C., K.P.J, and C.D.; manuscript writing and revision: B.M.B., I.J., R.B.W., S.E.B., D.H.C., K.P.J, and C.D.

## Competing interests

The authors declare no competing interests.
