## [Peer Review File · Nature Communications]

Stochasticity, determinism, and contingency shape genome evolution of endosymbiotic bacteriaREVIEWER COMMENTS

Reviewer #1 (Remarks to the Author):

In this manuscript by Boyd et al., the authors examined genome evolution of 34 lineages of endosymbiotic bacteria from *Columbicola* feather lice. They focused on the relative contributions of stochasticity, determinism, and contingency in shaping gene loss over time. I agree with the authors that the *Columbicola*-*Sodalis* symbiosis is potentially a great model for studying these processes. The manuscript is well-written and includes several novel results such as the data on the components of the respiratory chain. However, my main concern is that the novel read mapping methods established in this MS are not properly tested and that the phylogenetic methods used by the authors are suboptimal. These two issues should be addressed since they have the potential to strongly affect the results and the overall conclusions.

Major comments

(1) Novel pipeline

The authors say that they developed a novel pipeline for analyzing endosymbiont gene loss [<https://github.com/Ian-N-James/RepeatedAcquisitionOfEndosymbiontsInFeatherLice>]. However, this unpublished pipeline does not meet quality standards for a bioinformatics software. It does not contain sufficient documentation. There is no benchmarking or comparisons with standard tools. Most of the Processing scripts contain hard-coded information specific for this manuscript. As presented now, it is a set of custom scripts, not a fully reproducible pipeline. I understand that using standard metagenomics methods was difficult with the system since the sequencing coverage for many of the species was poor and the symbiont genomes were highly fragmented. However, the authors did not convince me that they properly tested their new pipeline, i.e., the new pipeline could be simply not showing the poor data quality.

Line 622: Validation of alignment/annotation approaches

The authors compared their novel approach to a metagenome assembly from a single species in which they suspected high endosymbiont coverage. In my view, such a comparison is not sufficient. Since the authors are presenting a novel way of analyzing endosymbiont data, metagenome assembly and genome binning should be carried out for all the samples. The authors should then compare the results, considering the sequencing coverage, to make it clear that their pipeline is superior. Also, that no genome assemblies were generated by this manuscript will limit further comparisons across systems and even highly fragmented assemblies are better than no assemblies.

Do any of the *Columbicola* lice house two or more *Sodalis* symbionts? Was there any cross-contamination among the samples? Are some of the predicted 'pseudogenes' resulting from low data quality/sequencing coverage? Such issues are easy to spot from metagenome assemblies. However, I'm not sure if the newly developed pipeline would distinguish these issues since the authors manually optimized the mapping criteria for every read set (Line 557).

(2) Phylogenetic inference

Fig 1. The phylogenetic tree inferred from 13 single-copy genes is extremely poorly resolved and contrasts with the high-quality louse phylogeny inferred from 977 genes. Drawing any conclusions about the symbiont origins and replacements from this tree is very preliminary. For resolving the relationship among short-branched taxa, more genes are needed, including faster evolving genes (and protein evolutionary models). Its taxon-sampling is also surprisingly poor. There are many *Sodalis* lineages missing from this tree. Not including *Sodalis* taxa from other insects (or *S. ligni* from wood) inevitably makes it look like the symbionts always originate from *Sodalis praecaptivus* HS1-like ancestors.

The authors further inferred a tree from 298 orthologs (Fig 1b), but for an unclear reason, only under a nucleotide evolutionary model (GTR+G) and for a subset of symbionts. This is not sufficient to draw conclusions about the symbiont evolution.

Is there any hard evidence that *S. praecaptivus*-like lineage is the ancestor of every *Sodalis* symbiont in *Columbicola* lice? Since the current sampling of 'free-living' *Sodalis*-allied lineages is extremely poor and *Sodalis* can also move horizontally across diverse insects, my guess would be that it's quite unlikely. The facultative *Sodalis* ancestors thus enter the symbiosis with *Columbicola* with different starting gene sets. Using HS1 as a reference is an oversimplification ignoring the pangenome present across different *Sodalis* lineages. I'd recommend the authors to reconstruct the pangenome of all *Sodalis* lineages from *Columbicola* metagenome assemblies to identify all accessory genes potentially missing from the core HS1 genome analyzed only by read mapping.

Minor comments

Please use color-blind friendly colors. Especially the combination of cyan, black, and magenta in Fig 3a is hard to read.

Line 24: We find that the process of genome degeneration [please add: in this system] is largely deterministic.

Line 34: Animals and plants – replace with Eukaryotes

Line 37: algae and plants – replace with the ancestor of algae and plants

Line 40: the cited papers mostly include animals, not 'many eukaryotic organisms'

Line 58: In contrast, some mutualistic insect-microbe – replace some with many?

Line 102: four different proteobacterial lineages: only three are clearly visible on the tree

Line 112-112: *Sodalis praecaptivus* is only on Fig 1a, its branch length information is not visible on Fig 1b where more genes were used.

Line 175. Please add a figure/table reference where to find the numbers of translation genes.

Line 319: with identical lifestyles, providing – add likely providing B-vitamins

Line 514: where -> were

Line 618: were the compared -> then

Line 653: I don't see how the minimal gene is set relevant here since the host, symbiont, and the host diet are all different from *Columbicola*.

Reviewer #2 (Remarks to the Author):

This paper reports analyses of genomic structures and gene inventories of 34 lineages of *Sodalis*-allied endosymbionts residing in feather-feeding bird lice of the genus *Columbicola*. The results appeared to suggest that the process of genome degradation is largely deterministic (convergent) due to the nutritional requirements of the host diet. However, genes with redundant functions seemed to have been randomly lost among lineages. Although the obtained results are not very novel or surprising, the analysis potentially provides valuable information because the targeted system is unique in that the symbionts repeatedly originated from closely related free-living ancestors, providing an appropriate basis to study dynamic evolutionary processes underpinning symbiosis.

I find the topic interesting, but would like to see more explanations in some parts of the paper.

L17-31: The summary looks somewhat too simple. I believe the authors can add more information as they have performed much more analyses.

L140, "plasmid sequence": What genes are encoded in the plasmid?

L213, "protein-rich diet": Protein-richness is not the point here. I believe "protein-rich" should be replaced with "vitamin-poor".

L223 and many other parts, "reciprocal": Sounds too ambiguous at least to me. Is this a common word in this context?

L317-345: I believe it's better to use the past tense in the Conclusion section.

L359, "later": latter?

L371, "b": How was "Genome retained" calculated? If this was based on sequence similarities,

what was the threshold? Why are there more than 34 plots?

L372-3, "fraction ~ endosymbionts": It would be better to rephrase.

L414-5: I would like to see more explanations about this equation.

L454, "later": latter?

L472, "prfB": Should be placed after "One gene".

L477, "variation": Does this mean variation observed in a single symbiont lineage?

L589-610: I'm not sure if it's appropriate to identify candidate genes as intact or pseudogenized in this manner. (e.g. The authors noticed that the method misidentified mutL, but how about other cases?) I would like to see some more explanations.

L601, "Expectation-Maximization (E-M) analysis": First mentioned in the previous page (L587).

L634-5, "The resulting ~ duplications: Does this mean that other endosymbiont genomes were not manually inspected?

L639, IDE: Should be spelled out.

L653, "determine": Sounds too strong. I don't believe these (especially the former) can be "determined" only by the gene inventories of *P. humanus* and *Riesia*.

L672 "its closest relative": What was the host species?

RESPONSE TO REVIEWERS' COMMENTS

Reviewer #1 General comment:

In this manuscript by Boyd et al., the authors examined genome evolution of 34 lineages of endosymbiotic bacteria from *Columbicola* feather lice. They focused on the relative contributions of stochasticity, determinism, and contingency in shaping gene loss over time. I agree with the authors that the *Columbicola*-*Sodalis* symbiosis is potentially a great model for studying these processes. The manuscript is well-written and includes several novel results such as the data on the components of the respiratory chain. However, my main concern is that the novel read mapping methods established in this MS are not properly tested and that the phylogenetic methods used by the authors are suboptimal. These two issues should be addressed since they have the potential to strongly affect the results and the overall conclusions.

- Response: We thank the reviewer for the constructive comments. Reviewer 1 raised concerns regarding the methodology described in the manuscript. To address the concerns raised, we have conducted extensive additional analyses. We believe the additional work addresses the reviewer's concerns and supports the conclusion described in the original manuscript.

Reviewer #1 Major comment 1) Novel pipeline.

Comment: The authors say that they developed a novel pipeline for analyzing endosymbiont gene loss [<https://github.com/Ian-N-James/RepeatedAcquisitionOfEndosymbiontsInFeatherLice>]. However, this unpublished pipeline does not meet quality standards for a bioinformatics software. It does not contain sufficient documentation. There is no benchmarking or comparisons with standard tools. Most of the Processing scripts contain hard-coded information specific for this manuscript. As presented now, it is a set of custom scripts, not a fully reproducible pipeline. I understand that using standard metagenomics methods was difficult with the system since the sequencing coverage for many of the species was poor and the symbiont genomes were highly fragmented. However, the authors did not convince me that they properly tested their new pipeline, i.e., the new pipeline could be simply not showing the poor data quality.

- Response: We apologize for any confusion. Yes, the pipeline was designed specifically for this study and optimized for the best performance with the focal taxa. To clarify this point, we changed the description of the pipeline in the results to, "We developed a novel pipeline to assemble and annotate the genomes of feather louse endosymbionts". In addition, we have added comments to the scripts, which enable others to better interpret the methods and possibly adapt the methods to other studies. The reviewer's concern regarding the validation of the pipeline is addressed below.

Comment: Line 622: Validation of alignment/annotation approaches

The authors compared their novel approach to a metagenome assembly from a single species in which they suspected high endosymbiont coverage. In my view, such a comparison is not sufficient. Since the authors are presenting a novel way of analyzing endosymbiont data, metagenome assembly and genome binning should be carried out for all the samples. The authors should then compare the results, considering the sequencing coverage, to make it clear that their pipeline is superior. Also, that no genome assemblies were generated by this manuscript will limit

further comparisons across systems and even highly fragmented assemblies are better than no assemblies.

- Response: As indicated by the reviewer, the assembly of genomes from many endosymbiont lineages presents a vexing problem, due to the presence of repetitive mobile genetic elements. Fragmented metagenome assemblies would have presented a challenge for creating an accurate inventory of the genes present within endosymbiont genomes, such as those included in this study. However, we also agree that we could have done more to validate the performance of our methodology. To improve our ability to assess our customized genome assembly and annotation methods, we have completed metagenome assemblies for two additional symbionts. These two symbionts, along with the original metagenome assembly, provided us with an opportunity to examine performance of our pipeline under different conditions. The focal endosymbionts included *Columbicola claytoni*, which has the largest genome and contains many pseudogenes; 2) *Columbicola fradei*, which has the lowest overall sequencing depth and lowest ratio of symbiont to host reads, and 3) *Columbicola rodmani*, which is predicted to be the most distinct from *S. praecaptivus* in our phylogenomic results and has the greatest overall sequence divergence. Thus, these endosymbionts allowed us to examine performance under the extremes in our dataset, while permitting a detailed interrogation of each.
- **Results of comparisons:** *De novo* assemblies were completed using the same libraries used to construct the original assemblies (quality control and processing were described in the original manuscript). The *C. claytoni*, *C. fradei* and *C. rodmani* assemblies yielded 55,314, 17,650 and 35,211 contigs, with median sizes of 2,084 bp, 6,351 bp and 2,647 bp respectively. BLASTN and BLASTX searches identified 17, 18, and 13 contigs belonging to the endosymbionts. The contigs were aligned to the *S. praecaptivus* genome using Mauve and genes were manually identified. Exhaustive inspection of intact genes revealed that >99% of the intact genes identified using our custom process were also identified in the metagenomes. After exhaustively searching the *de novo* assemblies, we identified two functionally different and intact proteins, which were not identified by our pipeline. Collectively, the results provide compelling evidence of the accuracy of our approach.
- **Data availability:** The methods and results of metagenome assemblies and assembly-assembly comparisons are described in the Supplementary Materials of the revised manuscript (see detailed breakdown in Fig. S8) and contextualized in the results section of the amended manuscript (see section, “Pace and Drivers of Genome Decay”). Endosymbiont contigs identified within metagenomic assemblies have been deposited in the figshare repository alongside assembly data originally generated in this study.

Comment: Do any of the *Columbicola* lice house two or more *Sodalis* symbionts? Was there any cross-contamination among the samples? Are some of the predicted 'pseudogenes' resulting from low data quality/sequencing coverage? Such issues are easy to spot from metagenome assemblies. However, I'm not sure if the newly developed pipeline would distinguish these issues since the authors manually optimized the mapping criteria for every read set (Line 557).

- Response: When assembling orthologs for phylogenetic inference using 13 genes, we manually examined the local *de novo* assemblies for evidence of dual endosymbionts within a single host. The software used, aTRAM, can and will create multiple assemblies of the target region from additional genomes, when they are present. We did not find evidence of multiple

endosymbionts in one louse species or cross-contamination of endosymbiont reads between libraries. Likewise, we did not observe evidence of contamination or multiple endosymbionts through manual inspection of reads mapped to the *S. praecaptivus* genome, done using the Geneious alignment viewer. Furthermore, we imposed cutoffs for read libraries exhibiting low data quality/sequencing coverage, with the symbiont of *Columbicola fradei* having the lowest coverage, but still being above the threshold.

Reviewer #1 Major comment 2) Phylogenetic inference.

Comment: Fig 1. The phylogenetic tree inferred from 13 single-copy genes is extremely poorly resolved and contrasts with the high-quality louse phylogeny inferred from 977 genes. Drawing any conclusions about the symbiont origins and replacements from this tree is very preliminary. For resolving the relationship among short-branched taxa, more genes are needed, including faster evolving genes (and protein evolutionary models). Its taxon-sampling is also surprisingly poor. There are many *Sodalis* lineages missing from this tree. Not including *Sodalis* taxa from other insects (or *S. ligni* from wood) inevitably makes it look like the symbionts always originate from *Sodalis praecaptivus* HS1-like ancestors.

- Response: This reviewer's comment reveals that the rationale for our phylogenetic approach was not clearly stated in the manuscript. We apologize for this oversight and have made corrections to the manuscript. We include additional explanation below.
- **Rationale and resolution:** The 13 gene tree was designed to provide a survey of louse endosymbiont diversity. As the tree suggests, some endosymbionts belong to the genus *Sodalis*, while others were found to be members of the genera *Arsenophonus*, *Enterobacter*, and *Pantoea*. We agree with the reviewer that the family level relationships in our tree disagree with the current NCBI accepted taxonomic scheme, which is largely based on Adeolu et al. (2016. J Syst Evol Microbiol. 66:5575-5599). While our tree largely supported the accepted relationship between Enterobacteriaceae, Erwiniaceae, Pectobacteriaceae, Yersiniaceae, and Bruguievoracaceae (as determined by Adeolu et al. 2016 and Li et al. 2021. Curr Microbiol. 78:856-866), our analysis did not return Morganellaceae as sister to these families. Despite the difference in the familial level topological associations, we did find that species largely group by family, according to Adeolu et al. (2016) and Li et al. (2021). Therefore, we concluded that the 13 gene data set was sufficient to resolve the generic level relationships of *Columbicola* endosymbionts, provide a broad examination of endosymbiont diversity, and provide an initial examination of internal and external branch lengths within *Sodalis* (discussed further below), while minimizing gap characters in our alignment matrix given available data.
- **Taxon sampling in the phylogenetic survey:** We agree with the reviewer that additional *Sodalis* genomes have become available since we initiated this study. The inclusion of new *Sodalis* genomes in a tree with our louse symbionts would provide a critical test of the close relationships observed between *S. praecaptivus* and *Sodalis* endosymbionts from *Columbicola* in our phylogenetic inference. To address this concern, we conducted additional phylogenomic analyses described below, which include additional *Sodalis* species and related genera.
- **Host-symbiont comparisons:** Direct comparison of the louse and endosymbiont trees (e.g. comparative phylogenetic analysis and investigation of branch lengths) was reserved for the endosymbiont tree derived from the analysis of 297 genes. By focusing comparisons on the 297 gene tree, we compared *Sodalis* and *Columbicola* species using trees inferred with the maximum number of single-copy orthologs possible. We have made changes to lines 97-99,

and more specific figure references on line 115, to differentiate phylogenetic and phylogenomic analyses and reduce confusion.

- **Ortholog choice:** We agree that more genes, (when available) would be ideal, and we have done this in a major additional analysis of 241 ortholog genes with extensive ingroup and outgroup sampling (see “Additional phylogenomic analyses completed” below). However, it is unclear whether targeting “faster evolving genes” would improve phylogenetic results. Selecting the genes with the “fastest” rate of base substitutions would potentially involve genes that are saturated and may exacerbate effects of long branch attraction, particularly when endosymbionts are included in the analysis. Second, we are confused by the reviewer’s phrase “faster evolving genes (and protein evolutionary models).” Translation of DNA strings to amino acid strings would likely mask the observed differences when targeting genes with a higher rate of base substitutions. Below, we describe additional phylogenomic methods and results that we believe will satisfy the reviewer’s concerns regarding the 13 gene tree, and whose structure supports the hypothesis of multiple origins of symbioses within *Sodalis*. This additional dataset includes increasing the number of orthologs used for phylogenetic inference and the use of amino acid sequences for phylogenetic inference.

Comment: The authors further inferred a tree from 298 orthologs (Fig 1b), but for an unclear reason, only under a nucleotide evolutionary model (GTR+G) and for a subset of symbionts. This is not sufficient to draw conclusions about the symbiont evolution.

- Response: Again, we apologize for the lack of clarity as to our rationale. As we described above, the original 13 gene tree was used to survey endosymbiont diversity and determine if endosymbionts from diverse louse species (determined by taxonomy) clustered together or were dispersed throughout the tree. Ultimately, we wanted to generate the best possible tree representing *Sodalis* endosymbionts of *Columbicola* when examining phylogenetic patterns observed in simulations (given a scenario of repeated acquisition by lice) and comparing louse and endosymbiont trees. Thus, we developed a tree based on 297 genes for direct comparison with the louse tree, which was based on 977 genes, as described in the methods section. Endosymbionts that did not belong to the genus *Sodalis* (as determined using the 13 gene tree), were excluded from this 297 gene tree, as were endosymbionts that did not yield a complete genome. As described above, we have added details to the manuscript to address this issue.
- **Model selection:** In the additional phylogenomic analyses described below, we have now included a step of model selection as suggested by the reviewer. However, in general, the family of GTR models contains the most complex nucleotide models, with μ and base frequencies estimated for all possible changes. Recent studies have indicated that model selection for less complex models is typically unnecessary, because GTR models perform just as well as “selected” simpler models in phylogenetic analyses (Abadi et al. 2019. Nat Commun. 10:934). However, we also recognize that all time reversible models, such as GTR, have limitations when examining endosymbionts, given that endosymbionts have a reduced capacity to repair oxidatively damaged bases (discussed in the manuscript). In our previous works on Enterobacterales, trees inferred using both reversible and non-reversible models yielded similar topologies regarding the placement of endosymbionts. Despite what we believe is a sound rationale for using a GTR-G model, we conducted additional phylogenomic analyses to address the reviewer’s concerns, including model selection, which are described below.

- **Additional phylogenomic analyses completed:** As noted above, reviewer 1 expressed concerns regarding taxon selection, model selection, and locus selection in phylogenetic and phylogenomic analyses. To address these concerns, we extended the 297 single-copy-ortholog gene set used to examine phylogenetic relationships between *Sodalis* endosymbionts of *Columbicola*, to additional newly added taxa. The analysis provided us with a phylogenetic tree with sampling similar to that of the original 13 gene tree, with the addition of key *Sodalis* species and relatives identified by the reviewer, and inferred from a much larger number of genes. Thus, we were able to provide a critical test of phylogenetic hypotheses using additional loci and under additional DNA and amino acid substitution models. Methods and results of these new analyses are described in subsections a-i below and in the manuscript and Supplementary Methods.
 - a. **Taxon and locus selection:** The 13 genes selected for our original phylogenetic analysis were designed to provide a survey of louse endosymbionts and representative Enterobacteriaceae species. The loci selected for the analysis were chosen to provide the optimum of sequence continuity in the presence of incomplete genome sequences for some endosymbiont species. To increase locus and taxon sampling as suggested by the reviewer, we focused on the 297 core genes previously identified, attempting to identify orthologs in additional *Sodalis* species, other louse endosymbionts, and species representing multiple families within Enterobacteriaceae. However, we had to exclude some species included in the original analysis due to a lack of ortholog recovery in this larger gene set. We focused on *Sodalis* endosymbionts of *Columbicola* lice, additional *Sodalis* species and relatives (diverse endosymbiont species and wood-decaying species), *Arsenophonus* endosymbiont from *Campanulotes compar*, *Enterobacter*-like endosymbionts from *Columbicola* species, *Pantoea* endosymbiont from *Columbicola arnoldi*, and representative species from different families within Enterobacteriaceae (table 1 below). Unfortunately, we were not able to recover many of the orthologs from draft genome assembly of the *Pantoea* endosymbiont from *C. arnoldi* and it had to be excluded from further analysis. However, the sampling did allow us to identify the closest relative of *Sodalis* endosymbionts recovered from lice and also demonstrated the polyphyly of louse endosymbionts.
 - b. **Data sources:** When annotated genomes were available, whole predicted protein coding sets for all newly added taxa were downloaded from NCBI Genome Assembly database (<https://www.ncbi.nlm.nih.gov/assembly>). We favored CDS described in the RefSeq annotation, but accepted the GenBank annotation when the RefSeq annotation was not available. To sample critical taxa for which predicted CDS were not available, we downloaded either un-annotated contigs from NCBI or the raw sequence reads from the NCBI short-read archive (<https://www.ncbi.nlm.nih.gov/sra>). In the case where we obtained an assembly lacking an annotation, we generated an annotation using RAST (Aziz et al. 2008. BMC Genomics. 9:75; Overbeek et al. 2013. Nucl Acid Res. 42:D206-D214). When raw reads were utilized for whole genome assembly, we prepared the reads using fastp v0.23.2 and performed a *de novo* assembly using metaSPAdes v3.14.0 (Chen et al. 2018. Bioinf. 34:i884-i890; Bankevich et al. 2012. J Compu Biol. 19:455-477). We then identified candidate contigs belonging to the endosymbiont genome using NCBI BLASTx v2.10.0+, comparing to a custom database composed of representative Enterobacteriaceae (Altschul et al. 1990. J Mol

- Biol. 215:403-410). These putative endosymbiont contigs were then annotated using RAST or MiGa (Rodriguez-R et al. 2018. Nucl Acid Res. 46:W282-W288).
- c. Ortholog identification: OrthoFinder v2.3.14 was used to identify orthologous gene groups among the newly assembled and downloaded gene sets (Emms and Kelly 2019. Genome Biol. 20:238). From the resulting orthogroups, we identified each group that contained one of the 297 single-copy-orthologs used in our original *Sodalis* specific phylogenomic analysis. We then examined summary statistics, removing two newly downloaded species, due to poor representation in the focal 297 gene set. This left us with 48 newly added taxa, in addition to the 32 existing endosymbionts and *Sodalis praecaptivus* used in the targeted phylogenomic analysis of *Sodalis*. Next, we filtered the orthogroups, rejecting any individual species contribution to orthogroup that contained a paralog (i.e. one species contributed more than one locus). Finally, if four or more taxa were not represented in the filtered groups (i.e. they never contributed a gene or had their contribution removed due to paralogy), whole ortholog groups were rejected. This left us with 241 single-copy-orthologs, representing 95% or more of the taxa; effectively minimizing missing data, while providing many loci for phylogenetic estimation.
 - d. Alignment and supermatrix construction: Genes within orthogroups were then aligned as translated amino acid sequences using MUSCLE v3.8.31, and converted back to nucleotides (Edgar 2004. BMC Bioinf. 5:113.). Examination of the base frequencies in first, second, and third codon positions revealed that third positions were biased towards AT base pairs in endosymbiont species, but not in non-symbiotic species. Bias in base composition could lead to long-branch attraction between endosymbionts and possibly lead to clustering of all *Sodalis* endosymbionts from *Columbicola* together with other endosymbionts near *S. praecaptivus*. Thus, to reduce the likelihood of long-branch attraction, we excluded third codon positions for phylogenetic analysis using nucleotide sequences (figure 1 below). The remaining first and second codon position alignments were filtered using ClipKit v2.1.1 for each of the 241 genes and were clustered into 10 groups based on the base frequencies and alpha parameters using K-means clustering (RAxML v8.2.12 was used to estimate parameters, clustering done using R v4.2.2; Stamatakis 2014. Bioinf. 30:1312-1313; Wickett et al. 2014. Proc Natl Acad Sci USA. 111:E4859-E4868; Steenwyk et al. 2020. PLoS Biol. 18:e3001007), the same method employed previously in this study. The gene alignments were then merged to form a supermatrix and a partition was created based on gene assignment into one of the 10 groups.
 - e. Inference under a GTR model: RAxML was used to find the maximum likelihood tree given the merged matrix and partition scheme under a GTR+G model. Support for the tree topology was based on 100 bootstrap replicates, also completed using RAxML.
 - f. Inference under selected DNA models: To explore additional models of base substitutions with our nucleotide alignments, we conducted additional phylogenetic analyses using IQ-TREE v1.6.12 (Chernomor et al. 2016. Syst Biol. 65:997-1008; Kalyanamoorthy et al. 2017. Nat. Methods 14:587-589; Hoang et al. 2018. Mol Biol Evol. 35:518-522; Minh et al. 2020. Mol Biol Evol. 37:1530-1534). First, we used IQ-TREE to identify the optimal model for each of the 10 partitions using the “TESTONLY” option, which performs a “jModelTest” like model fitting for each partition. IQ-TREE selected complex time-reversible models for each partition, which

- have been documented in the data repository. The models were employed within IQ-TREE to find the best tree and perform 1000 ultrafast bootstrap replicates.
- g. Inference using amino acids: Finally, we implemented phylogenetic inference based on the amino acid sequences. We generated merged alignment with a partition representing each protein sequence. We then used the “TESTMERGE” and “recluster 10” functions (PartitionFinder-like process) to find the optimal partition and model fitting within IQ-TREE. We selected this search process because it was more similar to our nucleotide partition finding methods offered within IQ-TREE and was feasible on the timeline available for manuscript revision. For example, using the more comprehensive “MFP+MERGE” option (similar to ModelFinder), it was estimated to take ~300 days to complete the search given our amino acid supermatrix. We then used the optimal partition and model for each partition to find the best tree and assessed support with 1000 ultrafast bootstrap replicates.
 - h. Results: The maximum likelihood analysis conducted using RAxML supported a close relationship between *Sodalis praecaptivus* HS1 and the *Sodalis* endosymbionts isolated from *Columbicola* species, even when wood-decaying and forest associated *Sodalis* species and relatives were included in the analysis (figure 2-4 below). This analysis also supported polyphyly of louse endosymbionts, arising from *Sodalis*, *Arsenophonus*, and *Enterobacter* progenitors. The branches most relevant to the key findings of our study were supported in 100% of bootstrap replicates under a GTR+G model (figure 2 below). IQ-TREE implemented with time-reversible models varying between partitions yielded a topology that was consistent with the RAxML tree obtained using the GTR+G model (figure 3 below). Again, the relevant topological arrangements were supported by 100% of bootstrap replicates. The consistency observed here between less complex models and parameter rich variations of the GTR model is consistent with the notion that GTR models perform just as well as “selected” models in phylogenetic analyses (Abadi et al. 2019). Likewise, phylogenetic analysis based on amino acid sequences with selected models yielded a topology consistent with the trees based on nucleic acids (figure 4 below). We conclude that the phylogenetic signal in the data is consistent, despite the use of different phylogenetic methods.
 - i. Data accessibility: Methods used in this additional analysis have been included in either the primary manuscript (see section “Repeated Acquisition of Heritable-intracellular Bacteria” and “Taxon Selection and Identification of Endosymbionts”) or Supplementary Materials (Fig. S2-S4; Supplementary Methods, Phylogenomics). Tree images have been added to the supplementary materials. Individual gene alignments, nucleotide and amino acid supermatrices, partition and character set files, and resulting phylogenetic trees have been deposited in the figshare data repository associated with this study.

Table 1. Samples used in phylogenomic analysis.

NCBI_identifier	Taxonomy	Host_or_habitat	Family	Insect_end	Annotatio
ASM434674v1	Sodalis ligni 159R	temperate forest soil	Bruguierivoraceaea	N	NCBI
ASM1686552v2	Sodalis ligni dw23	decomposing wood	Bruguierivoraceaea	N	NCBI
ASM1844957v1	Sodalis sp. dw_96	decomposing wood	Bruguierivoraceaea	N	NCBI
ASM434319v1	Biostraticola tofi	tufa biofilm	Bruguierivoraceaea	N	NCBI
ASM521745v1	Bruguirivorax albus	mangrove soil	Bruguierivoraceaea	N	NCBI
ASM51742v1	Sodalis praecpativus HS1	wound/crab apple impaled	Bruguierivoraceaea	N	NCBI
ASM3133291v1	Sodalis sp. BD_Bin2	Bathycollia distincta	Bruguierivoraceaea	Y	RAST
ASM1008v1	Sodalis glossinidius str. 'morsitans'	Glossina morsitans morsitans	Bruguierivoraceaea	Y	NCBI
SodNvir	Sodalis endosymbiont	Nezara viridula	Bruguierivoraceaea	Y	NCBI
ASM51740v1	Candidatus Sodalis pierantonius str.	Sitophilus oryzae	Bruguierivoraceaea	Y	NCBI
HBAv1	Sodalis baculum	Henestaris halophilus	Bruguierivoraceaea	Y	NCBI
ASM1877739v1	Candidatus Sodalis endolongispinus	Pseudococcus longispinus	Bruguierivoraceaea	Y	NCBI
ASM160262v1	Sodalis-like endosymbiont	Proechinophthirus fluctus	Bruguierivoraceaea	Y	NCBI
SoCistrobi_v1	Candidatus Sodalis sp. SoCistrobi	Cinera strobi	Bruguierivoraceaea	Y	NCBI
ASM2474857v1	Sodalis sp. Fle	Formica lemani	Bruguierivoraceaea	Y	NCBI
ASM2474859v1	Sodalis sp. Ffu	Formica fusca	Bruguierivoraceaea	Y	NCBI
ASM2474855v1	Sodalis sp. Fse	Formica selesi	Bruguierivoraceaea	Y	NCBI
ASM2464872v1	Sodalis sp. Ppy	Plagiolepis pygmaea	Bruguierivoraceaea	Y	NCBI
ASM2464874v1	Sodalis sp. Psp	Plagiolepis sp.	Bruguierivoraceaea	Y	NCBI
ASM187923v1	Sodalis sp. TME1	Llavenia axin axin	Bruguierivoraceaea	Y	NCBI
58635_F02	Pragia fontium	environmental	Budviciaceae	N	NCBI
ASM980092v1	Budvicia diplopodorum	Diplopoda sp.	Budviciaceae	N	NCBI
ASM229048v1	Shigella boydii	sewage water	Enterobacteriaceae	N	NCBI
ASM584v2	Escherichia coli str. K-12 substr. MG1655		Enterobacteriaceae	N	NCBI
ASM24018v2	Klebsiella pneumoniae subsp. pneumoniae HS11286		Enterobacteriaceae	N	NCBI
ASM703580v1	Enterobacter asburiae	river water	Enterobacteriaceae	N	NCBI
ASM1904710v1	Enterobacter cloacae	unknown	Enterobacteriaceae	N	NCBI
ASM1734891v1	Leclercia pneumoniae	human/sputum	Enterobacteriaceae	N	NCBI
ASM22467v1	Enterobacter soli	unknown	Enterobacteriaceae	N	NCBI
SRR3161958	Enterobacter symbiont	Columbicola koopae	Enterobacteriaceae	Y	MiGa
SRR3161914	Enterobacter symbiont	Columbicola macrourae 1	Enterobacteriaceae	Y	MiGa
SRR3161937	Enterobacter symbiont	Columbicola massoni	Enterobacteriaceae	Y	MiGa
SRR3161960	Enterobacter symbiont	Columbicola eowilsoni	Enterobacteriaceae	Y	MiGa
ASM479241v1	Pantoea vagans	eucalyptus	Erwiniaceae	N	NCBI
ASM2618v1	Erwinia tasmaniensis Et1/99	apple/pear tree	Erwiniaceae	N	NCBI
ASM1924377v1	Morganella morgani	Human/enteric	Morganellaceae	N	NCBI
ASM19615v1	Photorhabdus laumondii subsp. laur	Nematode mutualist/insect	Morganellaceae	N	NCBI
ASM476852v1	Arsenophonus nasoniae FIN	Nasonia vitripennis	Morganellaceae	Y	NCBI
SRR5308113	Arsenophonus symbiont	Campanulotes compar	Morganellaceae	Y	RAST
ASM207325v2	Pasteurella multocida OG		OUT_GROUP	NA	NCBI
ASM93157v1	Haemophilus influenzae OG		OUT_GROUP	NA	NCBI
ASM326983v1	Lonsdalea quercina	Poplar tree	Pectobacteriaceae	N	NCBI
ASM229144v1	Brenneria goodwinii	Quercus	Pectobacteriaceae	N	NCBI
ASM281248v1	Dickeya fangzhongdai	Pyrus pyrifolia	Pectobacteriaceae	N	NCBI
ASM174218v1	Pectobacterium wasabiae CFBP 330	Eutrema japonicum	Pectobacteriaceae	N	NCBI
ASM1013153v1	Acerihabitans arbors	tree sap	Pectobacteriaceae	N	NCBI
ASM351616v1	Serratia marcescens	marine	Yersiniaceae	N	NCBI
ASM2127628v1	Rahnella victoriana	soil	Yersiniaceae	N	NCBI

Figure 1. GC composition by codon position in species used in phylogenomic analysis.

Figure 2. Maximum likelihood tree inferred from a partitioned nucleotide supermatrix using a GTR+G model. Red tips represent endosymbionts from *Columbicola* species. Light blue tip represents endosymbiont from *Campanulotes compar*.

Figure 3. Maximum likelihood tree inferred from a partitioned nucleotide supermatrix using best fit models. Red tips represent endosymbionts from *Columbicola* species. Light blue tip represents endosymbiont from *Campanulotes compar*.

Figure 4. Maximum likelihood tree inferred from a partitioned amino acid supermatrix using best fit models. Red tips represent endosymbionts from *Columbicola* species. Light blue tip represents endosymbiont from *Campanulotes compar*.

Comment: Is there any hard evidence that *S. praecaptivus*-like lineage is the ancestor of every *Sodalis* symbiont in *Columbicola* lice? Since the current sampling of ‘free-living’ *Sodalis*-allied lineages is extremely poor and *Sodalis* can also move horizontally across diverse insects, my guess would be that it’s quite unlikely. The facultative *Sodalis* ancestors thus enter the symbiosis with *Columbicola* with different starting gene sets. Using HS1 as a reference is an oversimplification ignoring the pangenome present across different *Sodalis* lineages. I’d recommend the authors to reconstruct the pangenome of all *Sodalis* lineages from *Columbicola* metagenome assemblies to identify all accessory genes potentially missing from the core HS1 genome analyzed only by read mapping.

- Response: The reviewer raises several concerns that are addressed individually below.
- **Progenitor:** We are not claiming that the extant *S. praecaptivus* genome sequence is a perfect match to the free-living progenitor(s) of the *Columbicola* endosymbionts. However, based on the phylogenomic analyses, it represents the closest known free-living relative and undoubtedly shares a high level of similarity in its gene inventory with these endosymbionts. In support of this hypothesis, we present results from additional phylogenomic analyses, described in response to previous comments above and additional discussion below. Moreover, what this means is that our analysis is somewhat conservative, in the sense that there may be additional genes in the symbionts that could be discovered to comprise additional examples of evolutionary contingency. However, aside from mobile DNA, we actually identify very few “orphan” protein-coding genes in the metagenomic assemblies of the *Columbicola claytoni*, *Columbicola fradei* and *Columbicola rodmani* symbiont genomes that could yield novel functionality (as discussed below).
- **Horizontal movement of *Sodalis*:** The authors are unaware of evidence to support the hypothesis that “*Sodalis* can also move horizontally across diverse insects.” We first conducted a literature search finding limited evidence of intraspecific horizontal transmission and we did not find evidence of interspecific transmission. One publication noted a lack of evidence for horizontal transmission in natural systems (Dale et al. 2001. Proc Natl Acad Sci USA 98:1883-1888). Tzuri et al. (2021. Microbial Ecol 81:818-827) observed horizontal transmission of *Sodalis* between parasitoids that occupied the same host species. In this case, super-parasitism provided a mechanism by which horizontal transmission could occur, but they showed successful horizontal transmission within a species, not across species. The case noted by Tzuri et al (2021) likely has little bearing on our study, as lice have no known parasitoids. In the case of lice, where the entire lifecycle is typically passed on the body of a single bird species, mechanisms for any horizontal transmission between species of lice that live on other bird species would be extremely limited. In addition, horizontal transmission would leave a distinctive pattern in the phylogenetic tree of two distantly related species of lice (or insect hosts) having closely related symbionts united by a long highly supported stem branch. In this case the donor lineage would be the very close relative of the recipient lineage and we would see this signature in the tree. Instead the louse *Sodalis* species have long terminal branches connected by very short internodal branches, a pattern inconsistent with horizontal transmission and predicted by repeated acquisition from a more slowly evolving free-living ancestor (see below).
- **Host range:** The presence of multiple bacterial species of the same genus being present in diverse insect species has been given as evidence of interspecific horizontal transmission in bacterial species other than *Sodalis*, and it might possibly be seen as evidence for such a

phenomenon in *Sodalis*. For example, this type of explanation has widely been applied to explain the host range of *Wolbachia* species. However, there are key differences between *Wolbachia* and *Sodalis* species isolated from diverse insect species. First, phylogenetic analysis of *Wolbachia* produces phylogenetic trees with different topological structure and branch lengths, when compared to *Sodalis*. In *Wolbachia*, phylogenetic analysis using a supermatrix approach to infer ML tree (similar to the methods used here), resulted in a well-supported tree with internal and external branch of similar lengths (Gerth et al. 2014. Nat Commun. 5:5117). In *Sodalis*, we find a different phylogenetic pattern, a “comb” or “star” like topology with short internal branches, relative to external branches. These patterns suggest different processes of diversification resulted in the observed phylogenetic patterns. The pattern we observed in our study of *Sodalis* was consistent with modeling of diversification by divergence of lineages via host-association from a non-symbiotic progenitor. Second, we would like to highlight genomic differences between *Wolbachia* and *Sodalis*. The number of predicted intact protein coding sequences varies less between *Wolbachia* species, than between *Sodalis* species. This difference is likely due to the origins of these bacteria in diverse insect species. Thus, the 1) polyphyly of louse endosymbionts, 2) closely related species of lice harboring endosymbionts of different genera, 3) distantly related lice harboring endosymbionts of the same clade, 4) lack of congruence between louse phylogeny and endosymbiont phylogeny, and 5) the comb-like phylogeny of *Sodalis* endosymbionts with long terminal branches all support the repeated acquisition hypothesis for the origin of louse *Sodalis* endosymbionts.

- **Pan-genome:** To address the question of a *Sodalis* pan-genome, we collated all sequences from the contigs derived from the assemblies of the *Columbicola claytoni*, *Columbicola fradei* and *Columbicola rodmani* endosymbionts that did not align to *S. praecaptivus*. We then subjected these sequences to BLAST analysis to determine their identities. The sequences were then categorized as components of either (1) fragments of DNA that were not aligned by Mauve, (2) mobile genetic elements comprising IS-elements or phage, or (3) two unique protein-coding genes of bacterial origin that are not represented in the *S. praecaptivus* genome (See Fig. S8 in the Supplementary Materials). Notably, very few intact genes in the latter category were identified (three in total from the three symbionts, one is redundant), indicating that the majority of “orphan” sequences in the louse endosymbionts are remnants of mobile genetic elements. Again, it is important to note that our goal of exploring the role of stochasticity and contingency in the evolution of the feather louse symbionts does not mandate that we have an exhaustive understanding of the pan-genomic composition of the louse endosymbionts and/or their free-living progenitors and that the information contained in our study can be regarded as a conservative assessment.

Reviewer #1 Minor comments

1) Please use color-blind friendly colors. Especially the combination of cyan, black, and magenta in Fig 3a is hard to read.

-Response: -Response: Color schemes have been updated. The cyan, black, and magenta color scheme has been updated to a green, purple, and black color scheme. Cyan and magenta color scheme has been updated to green and purple color scheme. Color schemes were derived from ColorBrewer 2.0 (<https://colorbrewer2.org/>). In cases where purple and green gradients were needed (Fig. 3a and Fig. 3b), the gradients were derived using an online “Lch and Lab colour and gradient picker” by David Johnstone, (<https://davidjohnstone.net/lch-lab-colour-gradient-picker>)

with the end points being colors from ColorBrewer 2.0. Lab color space gradients were then exported and used. To avoid reusing the same color scheme for differing data types the existing green purple color scheme for the tabs in Fig. 4c has been changed to a tan and teal color scheme. Finally, the red and blue coloring of arcs in Fig. 4c is now perceptually uniform, and uses color maps from CET (colorcet.com).

2) Line 24: We find that the process of genome degeneration [please add: in this system] is largely deterministic.

- Response: DONE

3) Line 34: Animals and plants – replace with Eukaryotes

- Response: DONE

4) Line 37: algae and plants – replace with the ancestor of algae and plants

- Response: DONE

5) Line 40: the cited papers mostly include animals, not ‘many eukaryotic organisms’

- Response: This has been revised to “animals”

6) Line 58: In contrast, some mutualistic insect-microbe – replace some with many?

- Response: We are not comfortable with this change because the timing of origin for many symbioses has not been determined and some date back 10s to 100s of Ma.

7) Line 102: four different proteobacterial lineages: only three are clearly visible on the tree

- Response: Our tree supports four origins, *Sodalis*, *Arsenophonus*, *Enterobacter*-like, and *Pantoea*. They are highlighted in either red or blue.

8) Line 112-112: *Sodalis praecaptivus* is only on Fig 1a, its branch length information is not visible on Fig 1b where more genes were used.

- Response: *S. praecaptivus* was included in the phylogenetic analysis, but excluded in host-endosymbiont tree comparisons (i.e. during the conversion of the tree to distance matrix) because it does not have a host and unpaired taxa must be removed before comparing trees. Consistent with the comparative method used, we generated a tanglegram, in which *S. praecaptivus* was pruned from the tree. *S. praecaptivus* is present in the supplementary data, including the tree file.

9) Line 175. Please add a figure/table reference where to find the numbers of translation genes.

- Response: An xlsx file has been added to the data repository that includes gene counts for all COG categories and a description of each category. These data has been mentioned within the manuscript.

10) Line 319: with identical lifestyles, providing – add likely providing B-vitamins

- Response: DONE

11) Line 514: where -> were

- Response: DONE

12) Line 618: were the compared -> then

- Response: DONE

13) Line 653: I don't see how the minimal gene is set relevant here since the host, symbiont, and the host diet are all different from *Columbicola*.

- Response: We agree that these are different systems, however *Riesia* offers the best possible expectation of gene conservation with regards to B-vitamin metabolism requirements of the host, given the potential for host complementation (particularly given that *Columbicola* and *Pediculus* are both members of Phthiraptera). Given the general lack of annotated endosymbiont genomes in lice, we are unaware of a more suitable comparison.

Reviewer #2 (Remarks to the Author):

Comment: This paper reports analyses of genomic structures and gene inventories of 34 lineages of *Sodalis*-allied endosymbionts residing in feather-feeding bird lice of the genus *Columbicola*. The results appeared to suggest that the process of genome degradation is largely deterministic (convergent) due to the nutritional requirements of the host diet. However, genes with redundant functions seemed to have been randomly lost among lineages. Although the obtained results are not very novel or surprising, the analysis potentially provides valuable information because the targeted system is unique in that the symbionts repeatedly originated from closely related free-living ancestors, providing an appropriate basis to study dynamic evolutionary processes underpinning symbiosis. I find the topic interesting, but would like to see more explanations in some parts of the paper.

- Response: We agree with the reviewer, that findings in our study fit certain expectations. For example, convergence in gene loss has been inferred when comparing previously sequence endosymbionts. However, we believe that the system described here and the findings made possible by the system are novel. Previous studies have had to rely on non-independent comparisons of species already in the same symbiotic relationship (i.e. resulting from host-endosymbiont cospeciation) or independent comparisons between very distantly related endosymbiont species or host species. The lack of independence or lack of consistency in other systems would prevent the identification of the differential patterns of gene loss observed here as they relate to natural selection and stochastic events. The conclusions reached here, by differentiating patterns of gene loss, could not be conclusively derived in other systems.

Comment: L17-31: The summary looks somewhat too simple. I believe the authors can add more information as they have performed much more analyses.

- Response: We agree with the reviewer, that it would be optimal to expand on details in the summary. However, in the draft submitted for review, the summary contained 175 words and Nature Communications has a limit of ~150 words. Therefore, we actually had to reduce the length for the revised draft and were unable to add additional details.

Comment: L140, "plasmid sequence": What genes are encoded in the plasmid?

- Response: This plasmid is ~450Kbp and contains 350 protein coding genes with diverse functions. The annotation information for this plasmid is provided in GenBank CP006570.

Comment: L213, "protein-rich diet": Protein-richness is not the point here. I believe "protein-rich" should be replaced with "vitamin-poor".

- Response: As the vitamin content of feathers has not been directly assessed, we cannot make the claim. The statement “protein-rich diet” is supported by previous research and the fact that feathers are comprised predominantly of keratin and is supported by previous research.

Comment: L223 and many other parts, "reciprocal": Sounds too ambiguous at least to me. Is this a common word in this context?

- Response: The word “reciprocal” is derived from Latin, meaning “returning” or “alternating.” This fits with the observation that certain genes have “alternating” patterns of retention. To address the concern, we have directly defined the term at its first use within the manuscript on the line 223.

Comment: L317-345: I believe it's better to use the past tense in the Conclusion section.

- Response: DONE

Comment: L359, "later": latter?

- Response: DONE

Comment: L371, "b": How was "Genome retained" calculated? If this was based on sequence similarities, what was the threshold? Why are there more than 34 plots?

- Response: The genome retained represents the fraction of the HS genome present in endosymbionts and does not consider sequence similarity. We have added language to clarify this point to the figure legend. There are 36 points, because two taxa were excluded from subsequent analysis after this plot was generated. Those two are the endosymbionts of *C. tasmaniensis* and *C. passerinae*.

Comment: L372-3, "fraction ~ endosymbionts": It would be better to rephrase.

- Response: We have added details describing this metric within the figure legend.

Comment: L414-5: I would like to see more explanations about this equation.

- Response: Explanation: In technical terms, $0.5E$ is the mean of two Shannon Entropies (having a maximum possible value of 1) multiplied by 0.5 so that it contributes to one half of the relationship strength value. The other half of the relationship strength value is derived from the Hamming distance between strings, which corresponds to the strength of the direct or reciprocal/alternating match. However, because Hamming distances can be used to determine the strength of both direct and reciprocal/alternating matches $\sqrt{(H - 17)^2}$ is used to transform values to a scale from 0 to 17, regardless of whether the relationship is direct or reciprocal/alternating. This explanation is now presented in the supplementary information section accompanied by a callout in the Fig. 4 legend.

Comment: L454, "later": latter?

- Response: DONE

Comment: L472, "prfB": Should be placed after "One gene".

- Response: DONE

Comment: L477, "variation": Does this mean variation observed in a single symbiont lineage?

- Response: This statement was referring to variation between lineages. This difference is most pronounced when comparing symbiont and non-symbiont lineages and was excluded to reduce the potential of long-branch attraction. We have updated the text to clarify, "between lineages."

Comment: L589-610: I'm not sure if it's appropriate to identify candidate genes as intact or pseudogenized in this manner. (e.g. The authors noticed that the method misidentified mutL, but how about other cases?) I would like to see some more explanations.

- Response: Response: MutL is a special case that has been investigated empirically. More broadly, the annotation of genes as intact or pseudogene is typically subject to human-inferential bias and the idiosyncrasies of parameter selection. In the absence of empirical validation, some level of error is inevitable in this process, as acknowledged in the manuscript. The methodology that we utilized for classification is mathematical in nature and is underpinned by the notion that conservation of function is directly related to conservation of protein sequence. It relies on the establishment of distributions of sequence conservation in intact genes and pseudogenes that can be distinguished by the E-M algorithm. The utilization of this approach is ratified by the inspection of those distributions, shown in Fig 3b, revealing clear distinctions.

Comment: L601, "Expectation-Maximization (E-M) analysis": First mentioned in the previous page (L587).

- Response: DONE

Comment: L634-5, "The resulting ~ duplications: Does this mean that other endosymbiont genomes were not manually inspected?"

- Response: Yes, we utilized manual inspection of the genome from the symbiont of *C. claytoni* to validate results. That is, *C. claytoni* was a test case. This process was not undertaken for each symbiont. However, we have now manually reviewed additional taxa, see response to reviewer #1 to further validate the methods.

Comment: L639, IDE: Should be spelled out.

- Response: DONE

Comment: L653, "determine": Sounds too strong. I don't believe these (especially the former) can be "determined" only by the gene inventories of *P. humanus* and *Riesia*.

- Response: We have revised the language, using "approximate" rather than "determine."

Comment: L672 "its closest relative": What was the host species?

- Response: We regret that the statement was unclear and have made changes to address this issue. The closest relative is a cryptic species identified in a previous publication using mitochondrial DNA p-distances and we have modified this section to clarify.

REVIEWERS' COMMENTS

Reviewer #1 (Remarks to the Author):

The authors have adequately addressed most of my comments and greatly improved the manuscript. I only have a few minor comments about the methodology and data availability.

(1) Validation of alignment/annotation approaches

Thank you for this analysis showing that in this system, the alignment method provides comparable results to metagenome binning.

(2) Data availability

The authors show that metagenome binning resulted in fewer than 20 endosymbiont contigs from all the test metagenomes, which is better than most *Sodalis* draft genomes currently available. Given these results, I insist that all 34 metagenomes should be assembled and the draft bins containing endosymbiont contigs deposited to NCBI. Not uploading the binned symbiont genomes to public databases (or only uploading to FigShare) makes these genomes invisible to Blast searches (and thus to many researchers who do not have the resources to create custom Blast databases).

(3) Phylogenetic methods

The authors improved most of the methodology. There were only a few misunderstandings I wish to clarify.

Faster evolving genes

- Apologies for not expressing this clearly. What I meant was that for a tree that includes some very short comb-like branches (with zero or near-zero branch length), adding some faster-evolving genes might provide the phylogenetic signal needed to resolve these relationships. I completely agree with the authors that a careful selection of orthologs with varied evolutionary rates is critical. For example, some very slowly evolving (but short) genes such as ribosomal proteins are, in my experience, not particularly useful phylogenetic markers for the longest symbiont branches.

Amino acid models

- The reason I recommended the authors to use amino acid models (such as LG, WAG, etc. with +I+F+G; not necessarily the very complex GTR amino acid model) is that they provide more reliable results (4 vs. 20 characters) for datasets with many long branches expected to be affected by long-branch attraction.

Horizontal movement of *Sodalis*

- This is a very interesting discussion and should be potentially included in the main text. I only partly agree with the authors that "The pattern we observed in our study of *Sodalis* was consistent with modeling of diversification by divergence of lineages via host-association from a non-symbiotic progenitor." but I fully agree that there are clear differences between *Wolbachia* and *Sodalis* (and other symbionts such as *Arsenophonus*, *Hamiltonella*, *Cardinium*, etc.). These could be mentioned in the discussion as many researchers in the field do not realize these differences.

Reviewer #2 (Remarks to the Author):

I'm satisfied with the revision.

Reviewer #2 (Remarks on code availability):

I've checked the files only partially, but it appears they provide enough information.

RESPONSE TO REVIEWERS' COMMENTS

Reviewer #1 (Remarks to the Author):

The authors have adequately addressed most of my comments and greatly improved the manuscript. I only have a few minor comments about the methodology and data availability.

(1) Validation of alignment/annotation approaches

Thank you for this analysis showing that in this system, the alignment method provides comparable results to metagenome binning.

Response: *We are glad that we were able to address the reviewer's concerns regarding genome assembly.*

(2) Data availability

The authors show that metagenome binning resulted in fewer than 20 endosymbiont contigs from all the test metagenomes, which is better than most Sodalite draft genomes currently available. Given these results, I insist that all 34 metagenomes should be assembled and the draft bins containing endosymbiont contigs deposited to NCBI. Not uploading the binned symbiont genomes to public databases (or only uploading to FigShare) makes these genomes invisible to Blast searches (and thus to many researchers who do not have the resources to create custom Blast databases).

Response: *To maximize accessibility we generated metagenome assemblies for all remaining lice used in our study. All contigs from each assembly sharing sequence identity with the *S. praecaptivus* whole genome sequence were identified and binned, yielding at least 1-Mb of sequence for each sample with the exception of *C. adamsi*, whose assembly yielded only a few symbiont contigs of small size due to low sequence depth. Therefore, symbiont contigs from all samples except *C. adamsi*, were uploaded (<https://www.ncbi.nlm.nih.gov/genome/>). WGS accession identifiers and associated information has been provided as a supplementary table.*

(3) Phylogenetic methods

The authors improved most of the methodology. There were only a few misunderstandings I wish to clarify.

Faster evolving genes

- Apologies for not expressing this clearly. What I meant was that for a tree that includes some very short comb-like branches (with zero or near-zero branch length), adding some faster-evolving genes might provide the phylogenetic signal needed to resolve these relationships. I completely agree with the authors that a careful selection of orthologs with varied evolutionary rates is critical. For example, some very slowly

evolving (but short) genes such as ribosomal proteins are, in my experience, not particularly useful phylogenetic markers for the longest symbiont branches.

Response: *We thank the reviewer for the clarification. However, we do not expect faster-evolving would provide additional information in a scenario where endosymbiont lineages have diverged from an asymbiotic ancestor. In such a scenario, with rapidly evolving endosymbionts diverging from a slower evolving free-living parent population, differences in the rate of substitution/site/time between genes within endosymbiont lineages are likely less than the difference within orthologs when comparing symbiotic and asymbiotic lineages. In silico simulations suggest the evolutionary scenario describe will results in comb-like topologies where the bifurcations do not represent true relationships, instead are a limitation of tree inference (relationships emerging from host-endosymbiont cospeciation being the exception; Smith et al. 2019; BMC Evol Biol 13:109). We believe the additional analyses provided in our previous response already shows that the phylogenetic pattern seen here are consistent to data type and model parameters.*

Amino acid models

- The reason I recommended the authors to use amino acid models (such as LG, WAG, etc. with +I+F+G; not necessarily the very complex GTR amino acid model) is that they provide more reliable results (4 vs. 20 characters) for datasets with many long branches expected to be affected by long-branch attraction.

Response: *We thank the reviewer for clarification. Our previous results show that trees inferred using both nucleotides and amino acids are consistent and multiple amino acid models were considered in our analysis.*

Horizontal movement of Sodalis

- This is a very interesting discussion and should be potentially included in the main text. I only partly agree with the authors that “The pattern we observed in our study of Sodalis was consistent with modeling of diversification by divergence of lineages via host-association from a non-symbiotic progenitor.” but I fully agree that there are clear differences between Wolbachia and Sodalis (and other symbionts such as Arsenophonus, Hamiltonella, Cardinium, etc.). These could be mentioned in the discussion as many researchers in the field do not realize these differences.

Response: *We thank the reviewer for the comment and are happy that we could clarify the relevant phylogenetic arguments. The results of simulations were described in detail within an earlier publication (Smith et al. 2019; BMC Evol Biol 13:109) and are summarized in the discussion of the original manuscript. We are not sure what parameters of the simulation the reviewer disputes, but reviewer does highlight an area of future work. We agree with the reviewer, that a comparative review of Sodalis and Wolbachia phylogenetic patterns and genomic diversity is warranted, as both would highlight key differences. However, we believe these points fall outside the scope of this work and would be better served in a future publication devoted to the comparison.*

Reviewer #2 (Remarks to the Author):

I'm satisfied with the revision.

Response: *We are glad that we were able to address the reviewer's concerns.*

Reviewer #2 (Remarks on code availability):

I've checked the files only partially, but it appears they provide enough information.

Response: *We are glad that we were able to address the reviewer's concerns as we have made every effort to make the raw data and data products available.*